# Molecular characterization of projection neuron subtypes in the mouse olfactory bulb

Sara Zeppilli[1,2], Tobias Ackels[3,4†], Robin Attey[1†], Nell Klimpert[1], Kimberly D Ritola[5], Stefan Boeing[6,7], Anton Crombach[8,9], Andreas T Schaefer[3,4*], Alexander Fleischmann[1,2*]

[1]Department of Neuroscience, Division of Biology and Medicine, and the Robert J. and Nancy D. Carney Institute for Brain Science, Brown University, Providence, United States; [2]Center for Interdisciplinary Research in Biology (CIRB), Collège de France, and CNRS UMR 7241 and INSERM U1050, Paris, France; [3]The Francis Crick Institute, Sensory Circuits and Neurotechnology Laboratory, London, United Kingdom; [4]Department of Neuroscience, Physiology & Pharmacology, University College London, London, United Kingdom; [5]Janelia Research Campus, Howard Hughes Medical Institute, Ashburn, United States; [6]The Francis Crick Institute, Bioinformatics and Biostatistics, London, United Kingdom; [7]The Francis Crick Institute, Scientific Computing - Digital Development Team, London, United Kingdom; [8]Inria Antenne Lyon La Doua, Villeurbanne, France; [9]Université de Lyon, INSA-Lyon, LIRIS, UMR 5205, Villeurbanne, France

*For correspondence:
andreas.schaefer@crick.ac.uk (ATS);
alexander_fleischmann@brown.edu (AF)

†These authors contributed equally to this work

**Abstract** Projection neurons (PNs) in the mammalian olfactory bulb (OB) receive input from the nose and project to diverse cortical and subcortical areas. Morphological and physiological studies have highlighted functional heterogeneity, yet no molecular markers have been described that delineate PN subtypes. Here, we used viral injections into olfactory cortex and fluorescent nucleus sorting to enrich PNs for high-throughput single nucleus and bulk RNA deep sequencing. Transcriptome analysis and RNA in situ hybridization identified distinct mitral and tufted cell populations with characteristic transcription factor network topology, cell adhesion, and excitability-related gene expression. Finally, we describe a new computational approach for integrating bulk and snRNA-seq data and provide evidence that different mitral cell populations preferentially project to different target regions. Together, we have identified potential molecular and gene regulatory mechanisms underlying PN diversity and provide new molecular entry points into studying the diverse functional roles of mitral and tufted cell subtypes.

## Introduction

The mammalian olfactory system is unique among sensory systems in that it bypasses the thalamus: olfactory receptor neurons (ORNs) in the nose project to the olfactory bulb (OB), a forebrain structure containing – in the mouse – approximately 500,000 neurons per hemisphere (*Parrish-Aungst et al., 2007*). There, they synapse onto various interneurons and projection neurons. The latter directly project to a variety of cortical structures, including the anterior olfactory nucleus, piriform cortex, cortical amygdala, and the lateral entorhinal cortex (*Ghosh et al., 2011*; *Haberly and Price, 1977*; *Miyamichi et al., 2011*; *Sosulski et al., 2011*). This places OB projection neurons at a pivotal position to distribute processed olfactory information broadly across the brain.

Each ORN in the mouse expresses only one of approximately 1000 olfactory receptor genes (*Buck and Axel, 1991*; *Yoshihito, 2012*; *Zhang and Firestein, 2002*). ORNs expressing the same receptor project axons onto defined spherical structures, glomeruli (*Mori and Sakano, 2011*), containing a variety of neuropil including the apical dendrites of 10–50 projection neurons (*Bartel et al., 2015*; *Schwarz et al., 2018*). Historically, OB projection neurons have been divided into mitral and tufted cells (MCs, TCs), largely based on their soma location and dendritic and axonal projection pattern (*Figure 1—figure supplement 1*; *Haberly and Price, 1977*; *Imamura et al., 2020*; *Mori et al., 1983*; *Orona et al., 1984*): MC somata are located predominantly in a thin layer with their dendrites covering the deeper part of the OB external plexiform layer. Their axons project to a wide range of structures including posterior piriform cortex. TC axons, on the other hand, are restricted to more anterior forebrain structures and their cell bodies are distributed across the external plexiform layer, with dendrites largely restricted to superficial layers. Within the TC population, subdivisions have been made into deep, middle, superficial, and external TCs, largely based on soma position. MCs on the other hand are often morphologically described as a largely homogeneous population. However, branching patterns of lateral dendrites as well as soma size and apical dendrite length might allow for further subdivision (*Mouradian and Scott, 1988*; *Orona et al., 1984*; *Schwarz et al., 2018*). Moreover, projection patterns might differ based on soma position along the dorsomedial–ventrolateral axis of the OB (*Inokuchi et al., 2017*; *Chen et al., 2021*).

In parallel to this morphological diversity, numerous studies have described physiological heterogeneity both as a result of differential inputs from granule cells onto TCs and MCs (*Christie et al., 2001*; *Ezeh et al., 1993*; *Geramita et al., 2016*; *Phillips et al., 2012*) as well as intrinsic excitability and glomerular wiring (*Burton and Urban, 2014*; *Gire et al., 2019*). Consequently, TCs respond more readily, with higher peak firing rates, and to lower odor concentration in vivo (*Griff et al., 2008*; *Kikuta et al., 2013*; *Nagayama et al., 2014*), and earlier in the respiration cycle compared to MCs (*Ackels et al., 2020*; *Fukunaga et al., 2012*; *Igarashi et al., 2012*; *Jordan et al., 2018*; *Phillips et al., 2012*).

Within the TC and MC populations, biophysical heterogeneity has been more difficult to tie to specific cell types or subtypes. MCs show diversity in biophysical properties that is thought to aid efficient encoding of stimulus-specific information and is, at least in part, experience dependent (*Angelo et al., 2012*; *Burton et al., 2012*; *Padmanabhan and Urban, 2010*; *Tripathy et al., 2013*). Both in vivo and in vitro recordings suggest that a subset of MCs show regular firing, while others show 'stuttering' behavior characterized by irregular action potential clusters (*Angelo et al., 2012*; *Balu et al., 2004*; *Bathellier et al., 2008*; *Buonviso et al., 2003*; *Carey and Wachowiak, 2011*; *Desmaisons et al., 1999*; *Fadool et al., 2011*; *Friedman and Strowbridge, 2000*; *Margrie and Schaefer, 2003*; *Padmanabhan and Urban, 2010*; *Schaefer et al., 2006*). While TCs are heterogeneous, with, for example, external TCs displaying prominent rhythmic bursting, driving the glomerular circuitry into long-lasting depolarizations in vitro (*De Saint Jan et al., 2009*; *Gire and Schoppa, 2009*; *Gire et al., 2019*; *Najac et al., 2011*), a systematic assessment of biophysical variety is lacking so far. Moreover, differential centrifugal input from cortical and subcortical structures might further amplify this overall heterogeneity both between MCs and TCs as well as potentially within those different classes (*Boyd et al., 2012*; *Kapoor et al., 2016*; *Markopoulos et al., 2012*; *Niedworok et al., 2012*; *Otazu et al., 2015*).

Thus, anatomical projection patterns, in vivo odor responses, and intrinsic properties are known to show substantial variability across different projection neurons. Systematic investigation of different projection neurons, however, has been hampered by a scarcity of specific molecular tools. Interneuron diversity, on the other hand, in general has received considerable attention with numerous studies including in the OB (*Parrish-Aungst et al., 2007*; *Tavakoli et al., 2018*), aiming to provide a systematic assessment of morphology, physiology, chemotype and the basis for genetic targeting of distinct types of interneurons. For OB projection neurons, however, only little information about chemotypes (*Kiyokage et al., 2010*) is available at this point: *Cdhr1* and *Tbx21 Haddad et al., 2013*; *Nagai et al., 2005* have been shown to be selectively expressed in a subset of OB projection neurons. CCK distinguishes a subset of TCs (superficial TCs, [*Liu and Shipley, 1994*; *Seroogy et al., 1985*; *Short and Wachowiak, 2019*; *Sun et al., 2020*]). Vasopressin-expressing cells might constitute a further subset of superficial TCs (*Lukas et al., 2019*), and recently, the *Lbhd2* gene has been used to obtain more specific genetic access to MCs (*Koldaeva et al., 2021*). Heterogeneous expression of both the GABAa receptor and voltage-gated potassium channel subunits have been

observed (*Padmanabhan and Urban, 2010*; *Panzanelli et al., 2005*), albeit not linked to specific cell types. Expression of axon guidance molecules such as *Nrp2* might further allow subdivision of projection neurons across the OB (*Inokuchi et al., 2017*).

Hence, while some molecular markers can be used to define specific subsets of projection neurons, this description is far from complete. A comprehensive molecular definition of projection neuron types would help to classify and collate existing biophysical, morphological, and physiological data and delineate the distinct output streams of the OB. Moreover, it would provide a platform upon which further focused experimental approaches could be tied.

Single-cell or single-nucleus RNA sequencing has been used effectively to map out cell types across a variety of brain areas (*Macosko et al., 2015*; *Zeisel et al., 2018*), including inhibitory interneurons in the mouse OB (*Tepe et al., 2018*). As M/TCs constitute only ~10% of all OB neurons, we decided to enrich for projection neurons for single-nucleus (sn)RNA-seq. We then combined snRNA-seq with bulk RNA deep seq as well as additional snRNA-seq for OB neurons projecting to different cortical areas, thereby allowing us to disentangle different projection neuron classes by target area. We found that indeed both MCs and TCs fall into several, separable types, defined by expression of both common and overlapping gene regulatory networks. This work will therefore provide a molecular entry point into disentangling the diversity of OB projection neurons and defining the functional roles of different MC/TC types.

## Results

### Single-nucleus RNA sequencing of olfactory bulb projection neurons distinguishes mitral and tufted cell types

To characterize the molecular diversity of OB projection neurons, we used viral targeting and fluorescence-activated nuclei sorting (FANS) to enrich for OB projection neurons and characterized their transcriptomes using single-nucleus RNA sequencing (snRNA-seq) (*Figure 1A*, *Figure 1—figure supplement 2A,B*).

First, we injected a retrogradely transported Adeno-Associated Virus expressing nuclear GFP (rAAV-retro-CAG-H2B-GFP; *Tervo et al., 2016*) into multiple sites along the antero-posterior axis of the olfactory cortex, specifically into the anterior olfactory nucleus (AON) and piriform cortex (PCx) (*Figure 1A*). Histological analysis revealed that virus injections resulted in GFP expression in a heterogeneous population of OB cells labelling cells in the mitral cell, external plexiform, glomerular, and granule layers (*Figure 1B*). Sparse GFP expression in putative periglomerular and granule cells may have resulted from viral infection of migrating immature neurons from the rostral migratory stream or from diffusion of the virus from the injection site.

We dissected the olfactory bulbs of three injected mice, generated three independent replicates of single-nucleus suspensions, enriched for GFP expression using FANS (*Figure 1—figure supplement 2*), and performed snRNA-seq using 10× Genomics technology (*Figure 1A*). We performed a detailed quality check of the individual replicates, then combined nuclei for downstream analyses (*Figure 2—figure supplement 1*). We analyzed a total of 31,703 nuclei, grouped in 22 clusters that were annotated post hoc based on the expression of established marker genes for excitatory and inhibitory neurons and glial cell populations (*Figure 2A–C*). We initially used the combinatorial expression patterns of glutamatergic and previously characterized M/T cell markers (*Vglut1*, *Vglut2*, *Vglut3*, *Tbx21*, *Cdhr1*, *Thy1)* to identify putative OB projection neurons, comprising 23.66% (n = 7504) of all nuclei.

Next, we further subclustered the selected profiles, resulting in a total of nine molecularly distinct subpopulations. To assign preliminary labels to each of these cell types, we used marker genes previously employed in functional or single cell RNA-seq studies (*Nagayama et al., 2014*; *Tepe et al., 2018*). We also used available RNA in situ hybridization data from the Allen Institute for Brain Science to corroborate our preliminary assignments (*Figure 3—figure supplements 1* and *2*). Our analysis revealed eight molecularly distinct clusters of putative projection neurons and one cluster of putative periglomerular cells (*Figure 2D,E*). Among projection neurons, we identified three clusters of MCs (M1, M2, and M3) and five clusters of middle and external TCs (T1, ET1, ET2, ET3, and ET4).

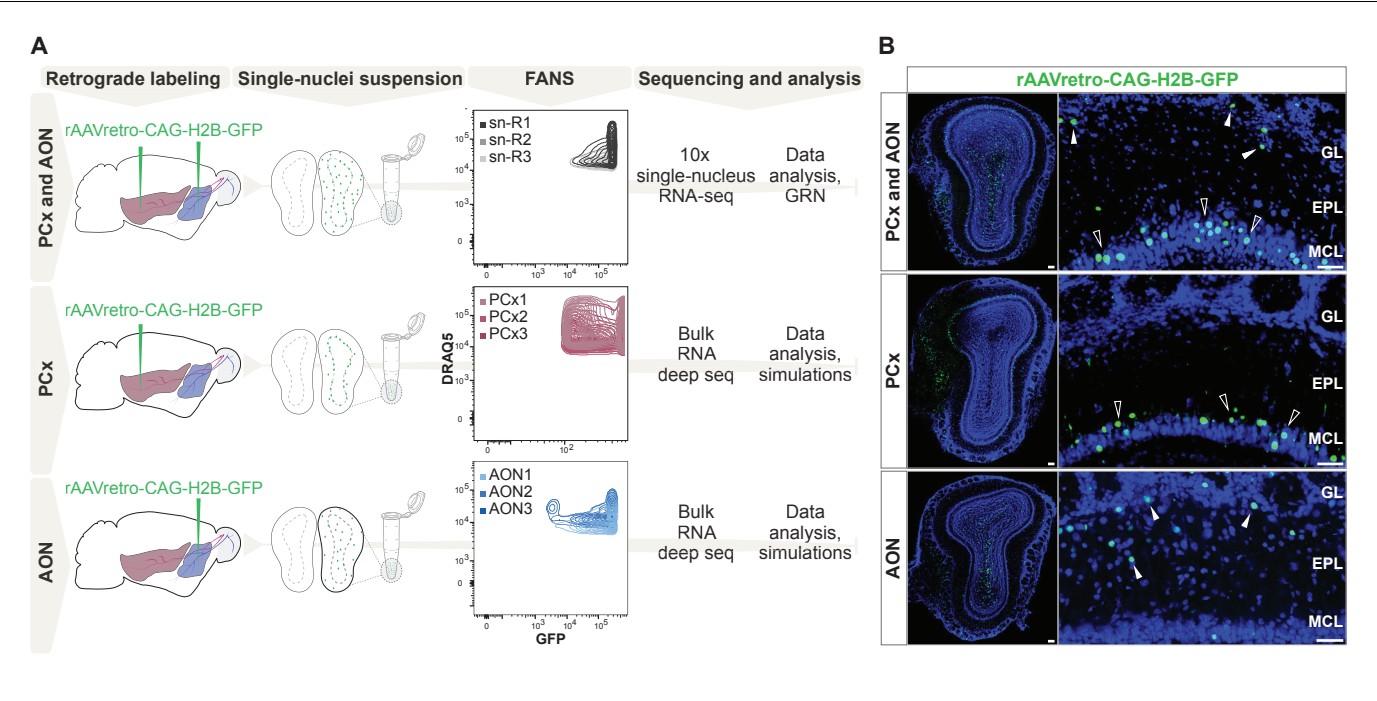

**Figure 1.** Comprehensive molecular profiling of olfactory bulb projection neurons. (**A**) Schematic representation of experimental design. Top: after injection of rAAVretro-CAG-H2B-GFP into PCx and AON, single nuclei were dissociated from three mice (single nuclei (sn) R1,2,3: replicates 1,2,3) and sorted using fluorescence-activated nuclei sorting (FANS). The population of nuclei is selected based on GFP and DRAQ5 (far-red fluorescent DNA dye). See *Figure 1—figure supplement 2* for detailed FANS plots. Sorted nuclei were sequenced using 10x single-nucleus RNA-seq. Middle and bottom: after injection of rAAVretro-CAG-H2B-GFP into PCx (middle) or AON (bottom), single nuclei were dissociated from three mice for each injection site and sorted using FANS (as described above and *Figure 1—figure supplement 2*). RNA extracted from sorted nuclei was prepared and sequenced using bulk RNA deep sequencing. PCx: Piriform Cortex; AON: Anterior Olfactory Nucleus; R: replicate; GRN: Gene Regulatory Network. (**B**) Representative coronal sections and high-magnification images showing GFP expression (in green) in the main olfactory bulb after injection of rAAVretro-CAG-H2B-GFP into PCx and AON (top), PCx only (middle), and AON only (bottom). Injection of the virus into PCx and AON resulted in GFP-expressing nuclei located in the mitral cell (empty arrowheads), external plexiform, glomerular (white arrowheads), and granule cell layers; injection into PCx resulted in GFP-expressing nuclei located in the mitral cell layer (empty arrowheads); injection into AON resulted in GFP-expressing nuclei located in the external plexiform and glomerular layers (white arrowheads) and granule cell layers. GL: glomerular layer; EPL: external plexiform layer; MCL: mitral cell layer; GCL: granule cell layer. Neurotrace counterstain in blue. Scale bars, 100 µm and 50 µm (high magnification).

The online version of this article includes the following figure supplement(s) for figure 1:

**Figure supplement 1.** Schematic representation of olfactory bulb cell types and their cortical projection targets.

**Figure supplement 2.** Enrichment of GFP-expressing nuclei using fluorescence-activated nuclei sorting (FANS).

## smFISH validates mitral and tufted cell types

To identify genes selectively expressed in OB projection neurons, we used the R-package glmGamPoi (*Ahlmann-Eltze and Huber, 2021*). We combined the average expression levels of the top differentially expressed genes for each cell type and found it to be highly specific for each cluster (*Figure 3A,C*). We then selected a few specific marker genes (*Figure 3B,D*) to validate projection neuron type identity by combining single-molecule fluorescent in situ hybridization (smFISH) with GFP staining upon rAAV-retro-CAG-H2B-GFP injection into the olfactory cortex.

We first investigated MC type-specific gene expression. Differential expression (DE) analysis identified the voltage-gated potassium channel *Kcng1*, the transcriptional regulator LIM homeobox 5 (*Lhx5*) and the serine-rich transmembrane domain 1 (*Sertm1*) as putative M1-specific marker genes. Two-color smFISH revealed extensive co-localization of *Kcng1* and *Lhx5* transcripts within the same subpopulation of cells in the MC layer (*Figure 3—figure supplement 1E,R*). Furthermore, *Kcng1*, *Lhx5*, and *Sertm1* expression was consistently observed in neurons expressing GFP (*Figure 3H*, *Figure 3—figure supplement 1B–D*). Next, DE analysis identified the mechanosensory ion channel

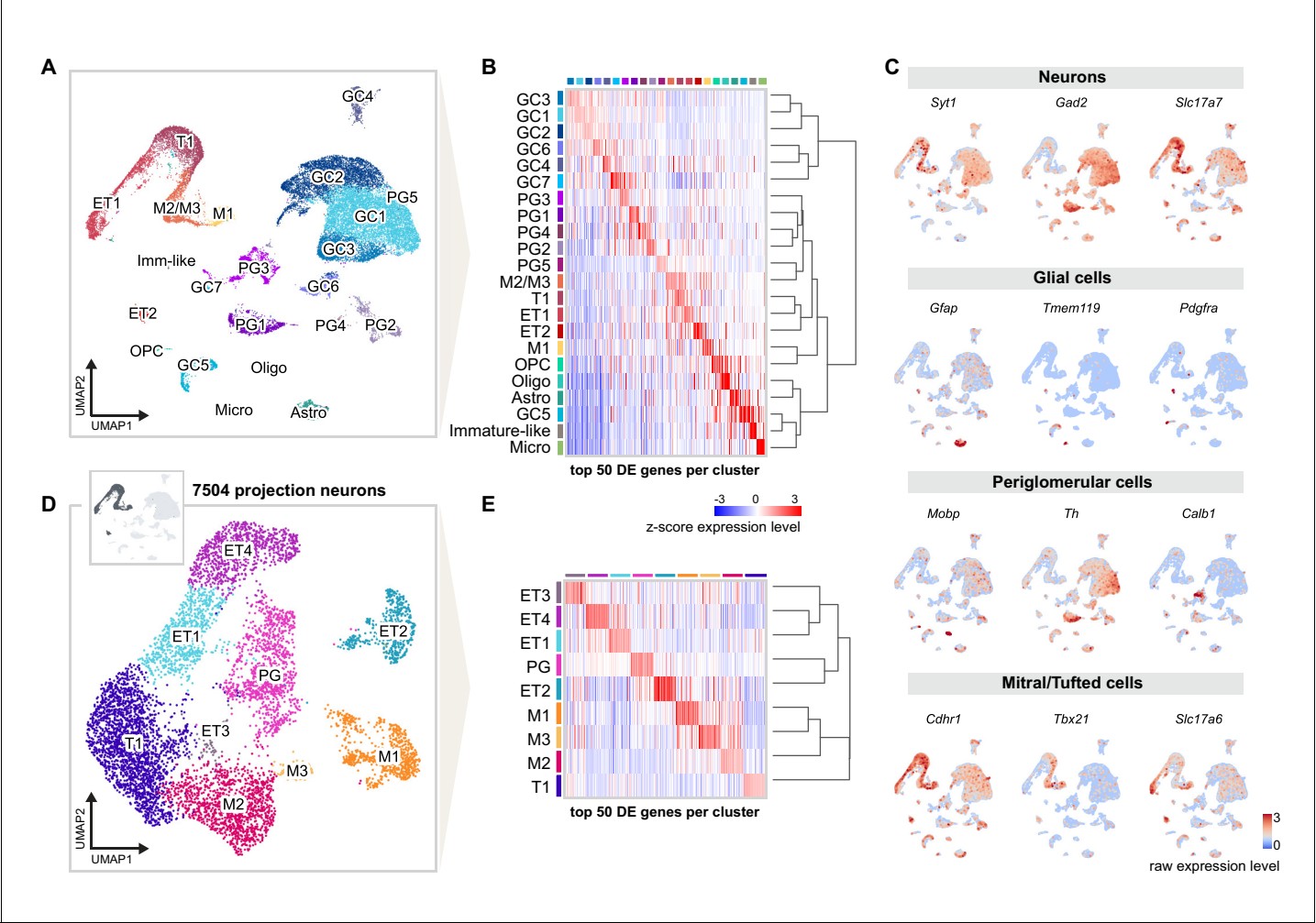

**Figure 2.** Single-nucleus RNA sequencing distinguishes distinct cell types and molecular signatures of OB projection neurons. (A) UMAP representation of gene expression profiles of 31,703 single nuclei combined from all replicates (R1, R2, R3) of mice injected into both AON and PCx, grouped into 22 clusters color-coded by cell type membership (GC: granule cell, PG: periglomerular cell, OPC: oligodendrocyte precursor cell, Micro: microglia, Astro: astrocyte, Oligo: oligodendrocyte, ET: external tufted cell, M: mitral cell, T: tufted cell, Imm-like: Immature-like cell). See *Figure 2— figure supplement 1* for detailed quality check of each replicate. (B) Matrix plot showing the z-scored expression levels of the top 50 differentially expressed (DE) genes for each cell population ordered by hierarchical relationships between distinct clusters. Each column represents the average expression level of a gene in a given cluster, color-coded by the UMAP cluster membership (from A). The dendrogram depicts the hierarchical relationships and is computed from the PCA representation of the data using Pearson correlation as distance measure and link by complete linkage. (C) UMAP representations of known marker genes for main cell populations (*Syt1*: neurons; *Gad2*: GABAergic neurons; *Slc17a7*: glutamatergic neurons; *Gfap, Tmem119, Pdgfra, Mobp*: glial cells; *Th, Calb1*: periglomerular neurons; *Cdhr1, Tbx21, Slc17a6*: mitral/tufted cells). Nuclei are color-coded by the raw expression level of each transcript. (D) UMAP representation of subclustering from the initial clusters M1, M2/M3, T1, ET1, and ET2 (cluster names from A), selected for the expression of known excitatory and mitral/tufted cell markers (shown in C), resulting in 7504 putative projection neurons grouped into nine distinct types. (E) Same matrix plot as described in (B) showing the z-scored expression levels of the top 50 DE genes for each projection neuron type ordered by hierarchical relationships and color-coded by the UMAP subcluster membership (from D).

The online version of this article includes the following figure supplement(s) for figure 2:

**Figure supplement 1.** Quality check of individual replicates of sn-RNA seq (sn-R1/R2/R3 dataset) shows the reliability of the data and the replicability of each cell type.

*Piezo2*, the transcription cofactor vestigial like family member 2 (*Vgll2*) and the zinc finger protein 114 (*Zfp114*) as putative M2-specific markers. Two-color smFISH revealed extensive co-localization of *Piezo2* and *Vgll2* transcripts within the same subpopulation of cells in the mitral cell layer (*Figure 3—figure supplement 1J,R*), and co-localization of M2-specific marker genes with GFP (*Figure 3I*, *Figure 3—figure supplement 1F,G*). Finally, we identified the calcium-dependent

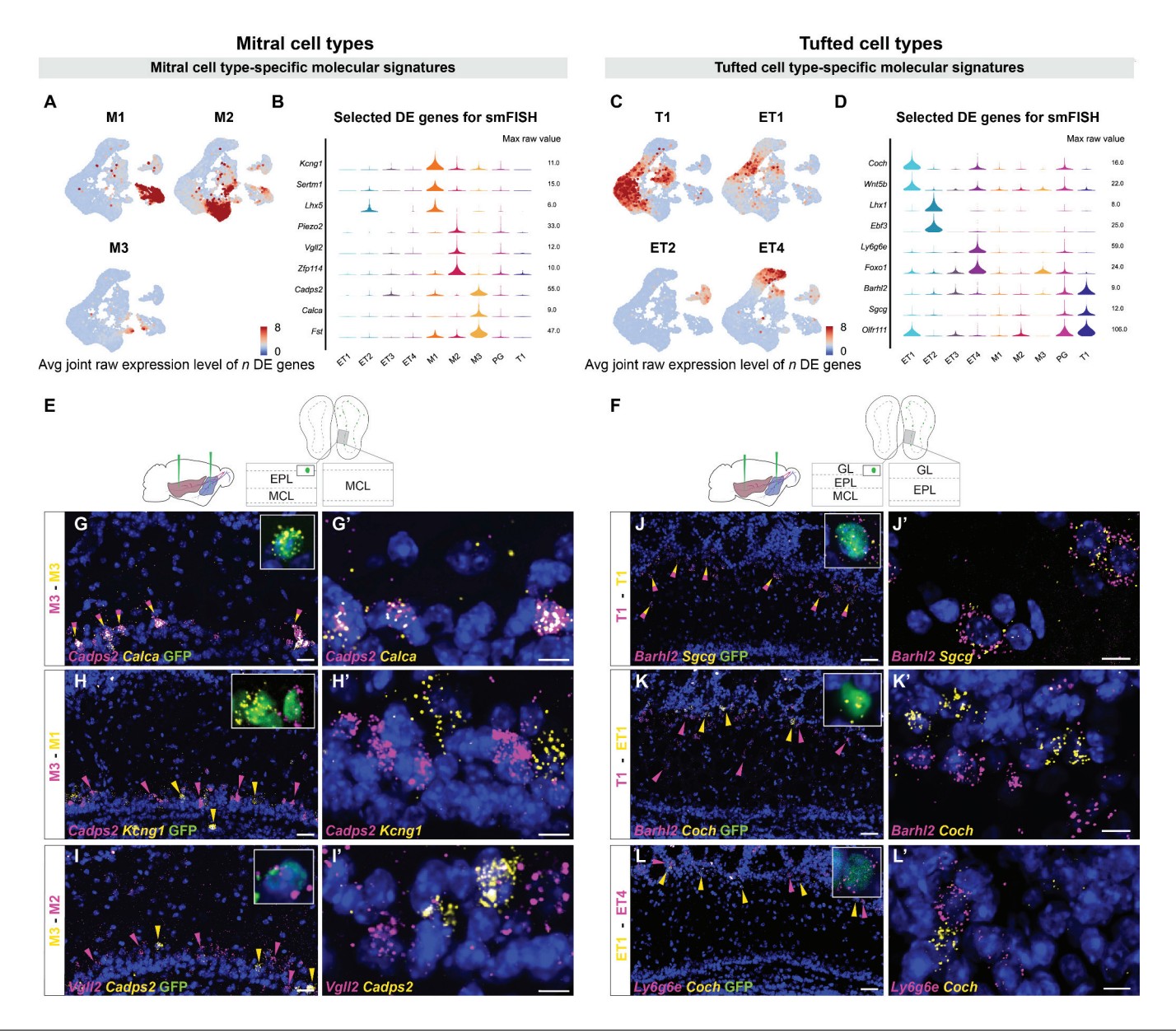

**Figure 3.** Histological validation of molecularly distinct mitral and tufted cell types. (**A**) Combined average (avg) raw expression level of the top *n* differentially expressed (DE) genes for each mitral cell type (M1 *n*=14, M2 *n*=11, M3 *n*=10), overlaid onto the subclustering UMAP space (shown in *Figure 2D*). DE genes were selected if their log fold change was greater than 4 (see Materials and methods for details). M1-specific genes: *Kcng1, Lhx5, Sertm1, Gabra2, Doc2b, Cntn6, Olfr1259, Nrp2, C1ql1, Ebf1, Baiap3, Adgrl2, Dsc2, Chrna5;* M2-specific genes: *Piezo2, Vgll2, Zfp114, Nts, Ros1, Samsn1, Grid2, Smpx, Itga4, Itga9, Sema6d;* M3-specific genes: *Cadps2, Calca, Fst, Ets1, Ednra, Cdkn1c, Mustn1, Smoc2, Cnr1, Ccno.* (**B**) Violin plots showing maximum raw expression value of selected mitral cell type-specific DE genes across mitral and tufted cell clusters for further validation with smFISH. (**C**) Combined average (avg) raw expression level of the top *n* DE genes for each tufted cell type (T1 *n*=9, ET1 *n*=7, ET2 *n*=9, ET4 *n*=6), overlaid onto the subclustering UMAP space (shown in *Figure 2D*). T1-specific genes: *Barhl2, Sgcg, Vdr, Olfr111, Olfr110, Cacna1g, Fam84b, Kcna10, Tspan10;* ET1-specific genes: *Coch, Wnt5b, Rorb, Chst9, Tpbgl, Clcf1, Rxfp1;* ET2-specific genes: *Lhx1, Ebf3, Trp73, Edn1, Ebf2, Nr2f2, Uncx, Psrc1, Dsp;* ET4-specific genes: *Ly6g6e, Foxo1, Siah3, Galnt12, Itga8, Ets2, Grik4.* (**D**) Violin plots showing maximum raw expression value of selected tufted cell type-specific DE genes across mitral and tufted cell clusters for further validation with smFISH. (**E, F**) Schematic representations of the smFISH images for validating projection neuron type-specific marker genes upon rAAVretro-CAG-H1B-GFP injection into PCx and AON. The schemes depict the laminar location visualized in the histological images from a coronal section of the ipsilateral hemisphere to the injection site. EPL: external plexiform layer; MCL: mitral cell layer; GL: glomerular layer. (**G–I**) smFISH showing combinatorial expression of mitral cell type-specific marker genes for M1, M2 and M3 cells in the mitral cell layer. High magnifications (top right) show co-labeling of viral GFP with the in situ mRNA probe. (**G and G'**). The M3 markers *Cadps2* and *Calca* are co-expressed in subpopulations of cells in the mitral cell layer, indicated by the yellow/magenta arrowheads. (**H and H'**) The M3

*Figure 3 continued on next page*

*Figure 3 continued*

marker *Cadps2* and M1 marker *Kcng1* are expressed in distinct subpopulations of cells in the mitral cell layer, indicated by the magenta and yellow arrowheads respectively. (I and I') The M3 marker *Cadps2* and M2 marker *Vgll2* are mutually exclusive in subpopulations of cells in the mitral cell layer, indicated by the yellow and magenta arrowheads respectively. For additional histological analysis and quantification of co-expression see *Figure 3—figure supplement 1*. (J–L) smFISH images showing combinatorial expression patterns of tufted cell type-specific marker genes for validating T1, ET1, ET2 and ET4 clusters as distinct projection neuron types in the external plexiform and glomerular layers. High magnifications (top right) show co-labeling of viral GFP with the in situ mRNA probe. As described for the mitral cell types, yellow or magenta arrowheads show non-overlapping patterns (K, K': T1–ET1 and L, L': ET1–ET4), and yellow/magenta arrowheads show co-expression patterns (J, J': T1–T1). For additional histological analysis and quantification of co-expression see *Figure 3—figure supplement 1*. DAPI counterstain in blue. Scale bars, 50 µm and 10 µm (high magnifications). The online version of this article includes the following figure supplement(s) for figure 3:

**Figure supplement 1.** Histological analysis of DE genes for distinct mitral cell types.

**Figure supplement 2.** Histological analysis of DE genes for distinct tufted cell types.

**Figure supplement 3.** Excitability-related, cell adhesion-related, and pan-OB projection neuron DE genes.

secretion activator 2 (*Cadps2)*, calcitonin (*Calca*), and follistatin (*Fst)* as putative M3-specific markers. smFISH revealed selective and extensive co-localization of *Cadps2* with *Calca* or with *Fst* transcripts within the same subpopulation of cells in the MC layer (*Figure 3G,G'*, *Figure 3—figure supplement 1N,R*). Furthermore, *Cadps2*, *Calca* and *Fst* expression was consistently observed in neurons expressing GFP (*Figure 3G,H*, *Figure 3—figure supplement 1K–M*). Importantly, two-color smFISH revealed that type-specific M1, M2, and M3 markers were expressed in largely non-overlapping populations of MCs: M1-specific *Kcng1* and *Lhx5* transcripts rarely co-localized with M2-specific *Vgll2* and *Piezo2* transcripts (*Figure 3—figure supplement 1O,P,R*); M1-specific *Kcng1* transcripts rarely co-localized with M3-specific *Cadps2* and *Calca* transcripts (*Figure 3H,H'*, *Figure 3—figure supplement 1Q,R*); M2-specific *Vgll2* transcripts rarely co-localized with M3-specific *Cadps2* transcripts (*Figure 3I,I'*, *Figure 3—figure supplement 1R*). Together, the selective co-localization of type-specific genes in non-overlapping populations in the mitral cell layer validates these three types as accurate and meaningful groupings of MCs, and their co-localization with GFP validates their identity as projection neurons.

DE analysis for TC type-specific genes identified the transcription factor BarH-like homeobox 2 (*Barhl2*), the gamma-sarcoglycan *Sgcg*, the vitamin D receptor (*Vdr*), and the olfactory receptors *Olfr110* and *Olfr111* as putative T1 markers. Two-color smFISH revealed extensive co-localization of *Barhl2* and *Sgcg* or *Olfr110/Olfr111* transcripts within the same subpopulation of cells in the external plexiform layer, indicative of middle tufted cells (*Figure 3J,J'*, *Figure 3—figure supplement 2B,L*). Furthermore, *Barhl2* and *Sgcg* expression was observed in neurons expressing GFP (*Figure 3J*). The coagulation factor C homolog (*Coch*) and the Wnt family member 5b (*Wnt5b*) were identified as putative ET1 markers. smFISH confirmed the expression of *Coch* and *Wnt5b* in a subpopulation of cells in the external plexiform and glomerular layers, although co-expression could not be recovered as consistently as for other markers (*Figure 3K*, *Figure 3—figure supplement 2D,L*). Moreover, *Coch* expression was observed in neurons expressing GFP (*Figure 3K*). The LIM homeobox 1 (*Lhx1*) and the early B-cell factor 3 (*Ebf3*) were identified as putative ET2 markers. Two-color smFISH revealed consistent *Lhx1* and *Ebf3* co-expression in a sparse subpopulation of cells at the boundary between the external plexiform and glomerular layers (*Figure 3—figure supplement 2F,L*), indicative of external TCs. Finally, we identified the lymphocyte antigen 6 family member 6GE (*Ly6g6e)* and the transcription factor Forkhead box O1 (*Foxo1*) as putative ET4 markers. smFISH revealed selective co-expression of *Ly6g6e* and *Foxo1* in a subpopulation of cells in the glomerular layer (*Figure 3—figure supplement 2H,L*), and co-localization of *Ly6g6e* with GFP validates their identity as external TCs (*Figure 3L*). Importantly, two-color smFISH revealed that type-specific tufted cell markers were expressed in largely non-overlapping populations of cells: ET1-specific *Coch* transcripts did not co-localize with T1-specific *Barhl2* transcript or with ET4-specific *Ly6g6e* transcripts (*Figure 3K,K',L, L'*, *Figure 3—figure supplement 2L*). Furthermore, ET4-specific *Foxo1* and *Ly6g6e* transcripts did not co-localize with T1-specific *Barhl2* transcript (*Figure 3—figure supplement 2I,L*), with ET2-specific *Ebf3* transcript (*Figure 3—figure supplement 2J,L*) or with ET1-specific *Wnt5b* transcript (*Figure 3—figure supplement 2K,L*). Overall, the selective co-localization of type-specific genes, their location within the olfactory bulb, their non-overlapping nature, and their co-localization

with GFP validate these five types of middle and external TCs as accurate and meaningful classifications.

## Inferring gene regulatory networks for projection neurons

The differential gene expression patterns revealed by transcriptome analysis are determined by the concerted action of transcription factors (TFs). We therefore set out to characterize cell types by their TF activity. To do so, we predicted regulatory relationships between TFs and target genes (that can be also TFs), taking into account independent genomic information about TF binding sites around the target gene-specific promoter. These gene regulatory networks are more robust against technical artifacts than the expression of individual genes, providing a complementary set of axes along which to cluster MCs and TCs. Ultimately, gene regulatory network analysis can yield more detail for classifying cell types and for understanding the molecular mechanisms that underlie their transcriptional differences.

To infer the regulatory networks of each type of projection neuron, we used the Single-Cell Regulatory Network Inference and Clustering pipeline (SCENIC, *Aibar et al., 2017*, *Van de Sande et al., 2020*). In brief, SCENIC consists of co-expression analysis, followed by TF binding motif enrichment analysis, and finally evaluation of the activity of regulons, that is a TF and its predicted target genes (*Figure 4A*). The result is a list of regulons and a matrix of all the single cells with their regulon activity scores (RAS, essentially an area-under-the-curve metric, see Materials and methods for details).

## Clustering on regulon activity corroborates molecular groupings of mitral and tufted cell types and allows further subdivision

We applied SCENIC to MCs and TCs (6484 cells), computing 86 regulons with a range of 8–551 target genes (median = 23.5). This greatly reduces the dimensionality of the data from >30,000 genes to 86 regulons, defining a new low-dimensional space in which to analyze relationships between nuclei. Using the regulon activity matrix, we computed a UMAP space and performed a Louvain clustering on the putative projection neurons (*Traag et al., 2019*; *Wolf et al., 2018*). We compared this gene regulatory network (GRN)-based clustering to the transcriptome-based clustering and observed that neighbor relationships are conserved between neurons (*Figure 4—figure supplement 1*). Indeed, several clusters in transcriptome space are clearly identifiable as clusters in GRN-space, and vice versa (e.g. ET2, ET4, M1+M3). However, we also noted significant differences between other clusters (ET1, T1, and M2). This disagreement suggests that network analysis provides orthogonal information relative to transcriptome analysis.

## Combinations of regulon modules characterize mitral and tufted cell subtypes

TFs activity is thought to be organized into coordinated network modules that determine cellular phenotypes (*Alexander et al., 2009*; *Irons and Monk, 2007*; *Suo et al., 2018*). To characterize how TFs are organized into such modules in OB projection neurons, we searched for combinatorial patterns of regulon activity. We used the Connection Specificity Index (CSI) to this end, which is an association index known to be suited for the detection of modules (*Fuxman Bass et al., 2013*; *Suo et al., 2018*). By computing the CSI on the basis of pairwise comparisons of regulon activity patterns across cells, we found that the 86 regulons grouped into seven modules (mod1–7) (*Figure 4B, C*).

Interestingly, regulon and module activity were not uniform within cell types. Rather, regulon activity suggested the existence of distinct subtypes within mitral and tufted cell types (*Figure 4D*). We used hierarchical clustering to further subdivide cell types, finding four subtypes of M1, three subtypes each of M2, T1, and ET2, and two subtypes of ET1 and ET4. To further investigate these subtypes, we used the modules to describe how the combined regulatory logic of distinct TFs contributes to the diversity of MC and TC subtypes. We asked if combinations of modules could uniquely describe the subtypes. To do so, we calculated the average module activity score per cell subtype. Next, we performed a hierarchical clustering on the subtypes (*Figure 4E*). We found that, when grouped by module activity, subtypes do not strictly group by type; rather, subtypes of different cell types tend to share similar module activity (*Figure 4E*). This is a consequence of the

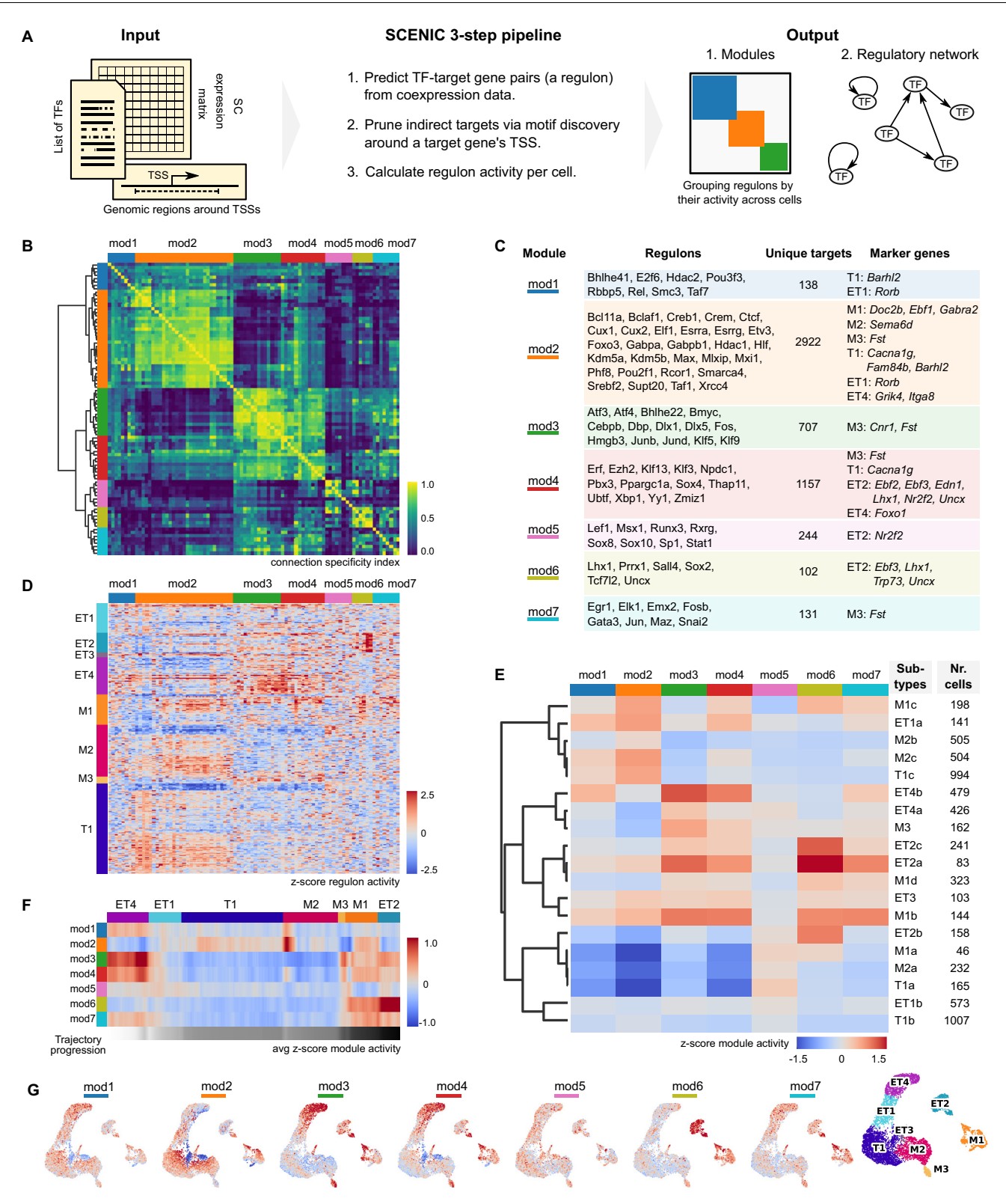

**Figure 4.** Mitral and tufted cell-specific regulons combine into modules. (**A**) Schematic representation of the network analysis pipeline, including the required input, the SCENIC protocol, and the output in the form of regulon modules and a regulatory network (in *Figure 5*). TSS: transcription start site; TF: transcription factor. (**B**) Hierarchical clustering of mitral and tufted cell-specific regulons using the Connection Specificity Index (CSI) as a distance measure results in seven modules (Ward linkage). The connection specificity index compares Pearson correlation coefficients (PCC) between

*Figure 4 continued*

regulons. If we take regulons A and B, their CSI is the number of PCC of A with other regulons and of B with other regulons that are lower than the PCC between A and B. This means that the CSI contextualizes a correlation between two regulons given how these regulons correlate to the rest. Prominent cross-module interactions are observed for mod1–2, mod3–4, and mod6–7. Module four showed interactions with all other modules. (C) Table listing the mitral and tufted cell-specific modules, their regulons, the number of unique target genes in each module, and cluster-specific marker genes found also as target genes in a given module. Marker genes were established by transcriptome analysis and shown in *Figure 3B and D*. (D) Regulon activity (columns) in mitral and tufted cell types (rows) defines subtypes within each cluster. Within each cell type, rows were ordered by hierarchical clustering (correlation distance, complete linkage). Columns clustered as in (B). (E) Module activity per cell subtype. Module activity is calculated as the average activity of its regulons for a given cell subtype. Rows were ordered by hierarchical clustering (correlation distance, complete linkage). Each cell subtype may be defined by a combination of active and inactive modules. For example, M2b and M2c are defined by relatively high activity in modules 1 and 2. (F) Quantification of changes in module activity through trajectory analysis. A single trajectory that traverses the cell types in the order ET4, ET1, T1, M2, M3, M1, ET2, is computed using PAGA (*Wolf et al., 2019*). Module activity along the trajectory is the average over a sliding window of 100 nuclei. Trajectory progression is depicted as a greyscale gradient from white to black. Along the same trajectory is also computed the regulon activity and expression levels of the corresponding TFs (*Figure 4—figure supplement 3*). (G) Module activity mapped on the projection neuron UMAP space (*Figure 2D*, rightmost UMAP for convenience). Color range as in (E).

The online version of this article includes the following figure supplement(s) for figure 4:

**Figure supplement 1.** Comparing transcriptomic and gene regulatory network-defined mitral and tufted cell types.
**Figure supplement 2.** Mitral and tufted cell type-specific marker genes found in regulons.
**Figure supplement 3.** PAGA-based trajectory analysis of mitral and tufted cell types.

differences between transcriptome- and GRN-based analysis: transcriptome clusters (cell types) are subdivided along the boundaries of GRN-based clusters. As we observed the grouping of subtypes along three neighboring clusters – ET1, T1, and M2 – we set out to better understand how module activity changes along these cell types. To this end, we used PAGA (PArtition-based Graph Abstraction) for trajectory analysis along the sequence of ET4, ET1, T1, M2, M3, M1, ET2 (*Figure 4F*, and for the corresponding UMAPs, see *Figure 4G*). We observed clear module activity gradients between ET4-ET1-T1, where modules 1, 3, 4, and 7 are high at the ET4 end of the cluster and low at T1. Module two forms an antagonistic gradient across ET1, high at T1 and low at ET4. Similarly, activity of module five changes in a gradient fashion from T1 to M2. Importantly, trajectory analysis not only detects gradients, but also discrete changes between cell types. For instance, the activity of module six shows a step-wise dynamic: high activity at ET2, medium at M1, and low for the other cell types. And module two has no activity in M3, while it is clearly present in its neighboring cell types M1 and M2.

## Regulon-based transcription factor networks reveal overlapping features of cell type identity

While TFs regulate a large number of target genes, central to cell identity are the interactions between them: TFs can regulate their own expression as well as the expression of other TFs, generating a TF network thought to be a core determinant of cell type identity (*Arendt et al., 2016*; *Becskei et al., 2001*; *Thieffry, 2007*). We thus looked at TF–TF interactions to visualize the TF network topology that defines MCs and TCs. We specifically asked whether MC and TC classes share common TF network features, or whether, as suggested by the analysis of genome-wide transcriptome and regulon analysis (*Figures 2–4*), MC and TC subtypes are defined by specific yet overlapping TF network features.

As a regulon is defined by a TF and a set of target genes, we constructed a (directed) network of TFs by taking from each regulon's target genes only the TFs that have regulons themselves (*Figure 5A*). Overall, we found one large set of interconnected TFs, one small component of two TFs, and 18 isolated TFs. Fifty-five of the 86 TFs (64%) show possible self-activating regulation, and several others form mutually activating pairs (e.g. *Lhx1* and *Uncx*). The network is dominated by three hub genes, two of which may self-activate: *Pbx3* (activates 17 TFs), *Bclaf1* (activates 11 TFs), and *Taf1* (activates 9 TFs). Their central position in this network suggests important roles in regulating transcription broadly across different cell types (e.g. Taf1, *Bclaf1*) or selectively in cell type-specific networks (e.g. Pbx3, *Figure 5—figure supplement 1*).

As anticipated from our module analysis, we find features that are shared across certain types of MCs and TCs rather than MC- or TC-specific network features (*Figure 5B,C*). For instance, we

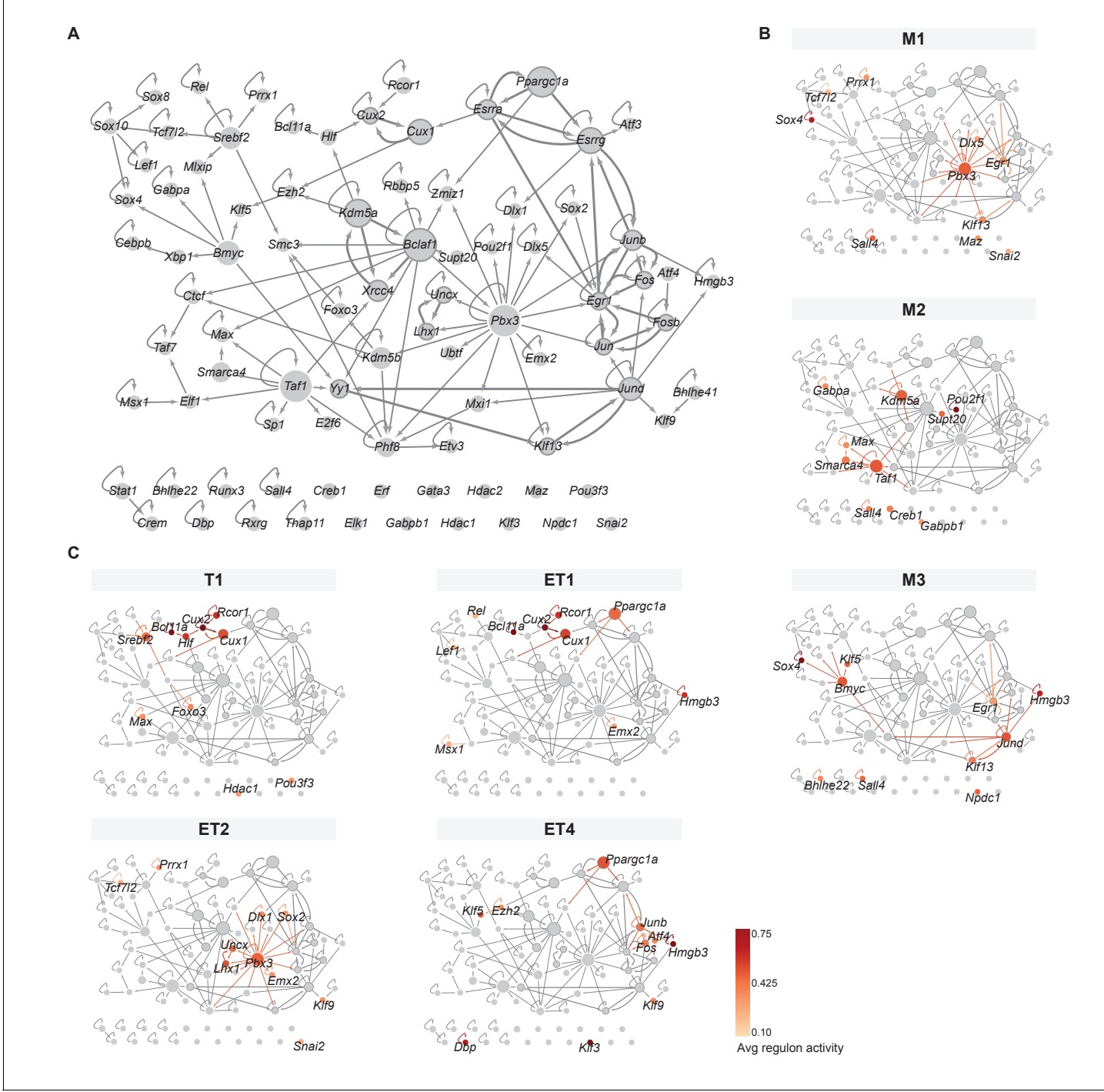

**Figure 5.** Mitral and tufted cell-specific transcription factor network derived from regulons. (**A**) Overview of mitral and tufted cell-specific transcription factor (TF) network, with node size scaled by the number of target genes and nodes colored with different shades of gray based on the outdegree (number of outgoing edges). Thick borders and edges denote cycles of 2 or three regulons. The three main hubs are: *Pbx3* (outdegree 17, target genes 545), *Bmyc* (outdegree 11, target genes 523) and *Taf1* (outdegree 9, target genes 551). (**B**) Same network as shown in A specific for mitral cell types (M1, M2, M3) with standardized regulon activity for the top 10 most specific regulons mapped onto the corresponding TF nodes (compare *Figure 4D,E*). Regulon specificity scores are shown in *Figure 5—figure supplement 1A*, with expression of the corresponding TFs visualized on UMAPs in *Figure 5—figure supplement 1B*. Mitral and tufted cell type-specific marker genes are visualized in the network in *Figure 5—figure supplement 2*. (**C**) Same network as shown in (**A**) specific for tufted cell types (T1, ET1, ET2, ET4) with standardized regulon activity for the top 10 most specific regulons. We omitted ET3 as it only has a few nuclei. As for mitral cell types, see also *Figure 5—figure supplements 1* and *2*.

The online version of this article includes the following figure supplement(s) for figure 5:

*Figure 5 continued*

**Figure supplement 1.** Top five mitral and tufted cell type-specific regulons.

**Figure supplement 2.** Mitral and tufted cell type-specific marker genes visualized as target genes in the gene regulatory network.

find that M1 and ET2 are characterized by the hub *Pbx3*. Yet both cell types are distinguishable by other TFs: M1 uses *Egr1* and *Dlx5* with *Pbx3*, whilst ET2 has a 2-cycle involving *Lhx1* and *Uncx*, which were both identified as marker genes specific for this cell type (*Figure 4—figure supplement 2*, *Figure 5—figure supplement 2*). Between T1, ET1, and ET4, we observe the gradients discussed above recapitulated in a transition between three groups of closely linked TFs. T1 is characterized by the close-knit group of Bcl11a, Cux1, Cux2, Hlf, and Rcor1, which are all members of module 2. Next, ET1 also employs these TFs, with the exception of Hlf and with the addition of Hmgb3 and Ppargc1a (modules 3 and 4, respectively). Interestingly, the estrogen receptors Esrra and Esrrg are in the top 15 of specific regulons for ET1, creating a target-gene-rich three-cycle with Ppargc1a: the second group of TFs. For ET4, Hmgb3 and Ppargc1a remain characteristic (but not Esrra and Esrrg), with Hmgb3 closely linked to the (third) group of Junb, Fos, and Atf4 (also members of modules 3 and 4).

Taken together, the analysis of TF regulatory networks suggests that individual MC and TC types share key TF network features, which might point towards common physiology or connectivity features. The differentially active TF network hubs and loops provide starting points for future investigation of the functional differences between the MC and TC types described here (*Figure 5—figure supplement 2*). Thus, the modules and the network serve as complementary approaches for studying cell type identity, with modules suited to classifying cells into types and subtypes and network analysis suited to investigating their functional differences.

## Simulating single-nucleus gene expression from bulk RNA deep sequencing

TCs preferentially target anterior olfactory regions, including the Anterior Olfactory Nucleus (AON), while MCs target anterior and posterior olfactory areas (*Imamura et al., 2020*). Therefore, we asked whether the genetic diversity within MC types could provide information about their projection targets. To investigate this question, we again injected rAAV-retro-CAG-H2B-GFP into the olfactory cortex, albeit now *either* into the AON *or* the posterior piriform cortex (pPCx) (*Figure 1A*). For each injection site and in three biological replicates, we then enriched for GFP expression using FANS (*Figure 1—figure supplement 2C–F*) and prepared RNA for bulk RNA deep sequencing (*Figure 1A*).

Bulk RNA sequencing represents molecular information from a variety of different cell types. Given that a substantial fraction of isolated nuclei in our experiments was comprised of granule and periglomerular cells, in addition to projection neurons, we devised a novel computational approach to recapitulate the constituent cell types from bulk RNA sequencing data by simulating single-nucleus expression profiles. Previous methods simulated the transcriptome of a single cell based on the overall distribution of gene expression levels in the bulk RNA sequencing data, producing many nuclei that were similar to each other and to the original bulk expression profile (*Konstantinides et al., 2018*; *Avila Cobos et al., 2018*; *Zhu et al., 2016*). This worked well for clean bulk RNA-seq datasets with only one cell type, but for our mixed datasets, the simulated nuclei resembled an unrealistic average of the constituent cell types. Therefore, to capture the diversity contained within our bulk RNA-seq datasets, we used regulons as the unit of analysis to create simulated nuclei with more biologically realistic transcriptomes (*Figure 6A,B*).

We first compared simulated and sn-R1/R2/R3 nuclei by using principal component analysis to project both sets of nuclei into a shared low-dimensional space. We used these principal components as input to a UMAP projection to visually inspect relationships between simulated and sn-R1/R2/R3 nuclei (*Figure 6B*, step 1). Consistent with histology and sn-R1/R2/R3 analyses (*Figures 1* and *2*), simulations from both AON-injection and pPCx-injection bulk RNA-seq datasets (AON-sim and PCx-sim, respectively) contained cell types other than projection neurons, and this contamination was more pronounced in the AON-sim dataset. The dispersion of simulated nuclei throughout this space indicated that simulations successfully recapitulated the diversity of cell types in the bulk

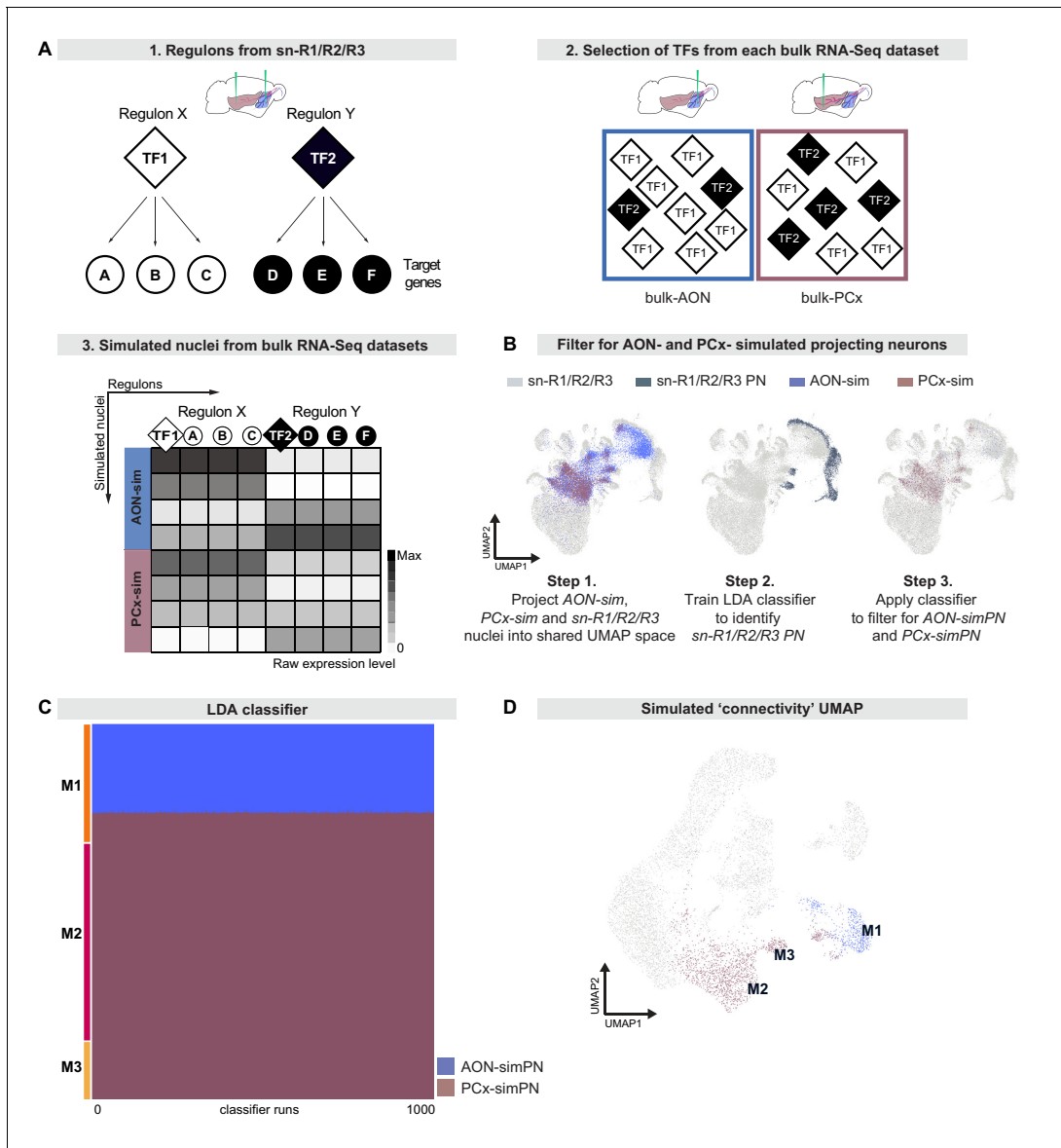

**Figure 6.** Regulon-based simulations from bulk RNA deep sequencing data suggest that mitral cell types have distinct projection targets. (**A**) Schematic representation of strategy to integrate bulk RNA-seq and single nucleus RNA-seq data. (A1) Simulations use regulons inferred from sn-R1/R2/R3 data. A regulon consists of a transcription factor (TF) and all target genes (A, B, etc) that are activated by that transcription factor. (A2) When simulating nuclei from bulk RNA-seq data, the probability of selecting a given regulon is determined by the abundance of its transcription factor in the bulk RNA-seq dataset. (A3) Nuclei are simulated from each bulk dataset (AON-sim and PCx-sim) through random sampling of regulons with replacement. This method maintains broad differences between datasets while accounting for heterogeneity within each dataset. (**B**) AON-sim, PCx-sim and sn-R1/R2/R3 nuclei projected into a shared UMAP representation. Step 1: Blue indicates AON-sim nuclei and purple indicates PCx-sim nuclei. Step 2: Darker color indicates sn-R1/R2/R3 projection neurons. Step 3: Blue indicates AON-projecting simulated projection neurons (AON-simPN) and purple indicates pPCx-projecting simulated projection neurons (PCx-simPN). (**C**) Linear Discriminant Analysis (LDA) classifiers were trained on AON-simPN and PCx-simPN, then used to predict the projection target of sn-R1/R2/R3 mitral cells to investigate projection targets of sn-R1/R2/R3 derived types. Each row represents one mitral cell. Each column represents one of 1000 LDA classifiers. Blue indicates that the mitral cell was classified as AON-simPN, and purple indicates that the mitral cell was classified as PCx-simPN. Within each mitral cell type, cells are sorted vertically by their predicted projection target. (**D**) UMAP representation color-coded by predicted projection target. Cells in blue were predicted to be AON-simPN by all 1000 classifiers. Cells in purple were predicted to be PCx-simPN by all 1000 classifiers.

RNA-seq datasets, as each cell class from the combined sn-R1/R2/R3 data had some simulated nuclei in its vicinity (*Figure 6B*, step 1). To account for this contamination and filter for only putative simulated projection neurons, we trained linear discriminant analysis (LDA) classifiers to predict whether sn-R1/R2/R3 nuclei were projection neurons based on the 30 top principal components that defined the shared low-dimensional space. These classifiers accurately and consistently classified projection neurons, with a mean Jaccard index (a measure of similarity between predicted and true labels) of 99.0% and a standard deviation of 0.07% over 1000 classifiers. We then applied this classifier to the AON-sim and PCx-sim nuclei, designating those simulations predicted to be projection neurons by all 1000 classifications as putative simulated AON- and PCx-projecting neurons (AON-simPN and PCx-simPN, respectively) (*Figure 6B*, step 3).

To directly compare AON-simPN and PCx-simPN to MCs characterized through sn-R1/R2/R3, we next used principal component analysis to define a shared low-dimensional space for MCs, AON-simPN and PCx-simPN only. To investigate potential differences in projection target between MC types, we trained 1000 LDA classifiers to predict the projection target of simulated projection neurons based on the 30 top principal components that defined this shared low-dimensional space (mean Jaccard index: 94.3%; standard deviation: 6.1%). We then used these classifiers to predict the projection targets of sn-R1/R2/R3 MCs. Interestingly, we consistently found different predicted targets for the molecularly defined MC types. All M2 and M3 MCs were classified as PCx-simPN by 100% of classifiers, suggesting that these mitral types preferentially project to posterior targets (*Figure 6C*). In contrast, 73.5% of M1 MCs were classified as AON-simPN by at least 80% of classifiers, suggesting that M1 MCs preferentially project to anterior targets (*Figure 6C*). These findings suggest that the molecular subcategorization of MCs may delineate differences in connectivity. The types M1, M2, and M3 were defined by gene expression and regulon activity, but these results suggest that they also describe projection target specificity. This correspondence between molecular identity and projection target is further demonstrated by the segregation of MCs by predicted projection target in a UMAP space defined by gene expression (*Figure 6D*).

## Targeted snRNA-seq validates predictions of preferential connectivity for molecularly defined mitral cell types

To validate our prediction that M2 MCs preferentially project to posterior targets and M1 MCs preferentially project to anterior targets, we performed a small volume rAAV-retro-CAG-H2B-GFP injection into the pPCx. While small volumes will likely result in a smaller number of labeled projection neurons overall, such focal injection will enhance specificity. We dissected the olfactory bulbs, generated single nuclei, enriched for GFP expression using FANS, and performed snRNA-seq using 10x Genomics technology (sn-PCx dataset) (*Figure 7A*). To assign cell type identity, we first identified projection neurons by transforming the expression matrix into the space of 30 principal components defined on the original, larger sn-R1/R2/R3 RNA-seq dataset. One thousand LDA classifiers, each trained and validated on the original sn-R1/R2/R3 dataset (mean jaccard index = 0.95), designated 57 out of 62 cells from the sn-PCx dataset as projection neurons (*Figure 7B*).

Next, we transformed the expression matrix of these projection neurons into a low-dimensional space defined by the expression of the 86 regulons from our SCENIC analysis on the original projection neurons sn-R1/R2/R3 dataset. These regulons create a low-dimensional space in which to compare this new, targeted sn-PCx dataset to our original dataset. To quantify this comparison, we used LDA classifiers, trained on the original mitral cells, to predict the cell type of these new projection neurons based on regulon activity scores. These classifiers were then applied to the regulon activity scores of the new projection neurons (*Figure 7C*).

Consistent with our prediction based on simulated transcriptomes, the vast majority (44 of 57) PCx-projecting mitral cells were classified as M2 cells by at least 80% of classifiers. In contrast, only 6 of 57 PCx-projecting mitral cells were classified as M1 cells (analysis of M3 projections was precluded by their low prevalence). Were M2 cells present in the sn-PCx population at the same rate as in the general mitral cell population, observing this many or more M2 cells would be unlikely (binomial test, p=0.001, see Materials and methods for details). Similarly, the scarcity of M1 neurons would be unlikely if M1 cells were present in the sn-PCx mitral population at the same rate as the general mitral population (binomial test, p=5.57e-05).

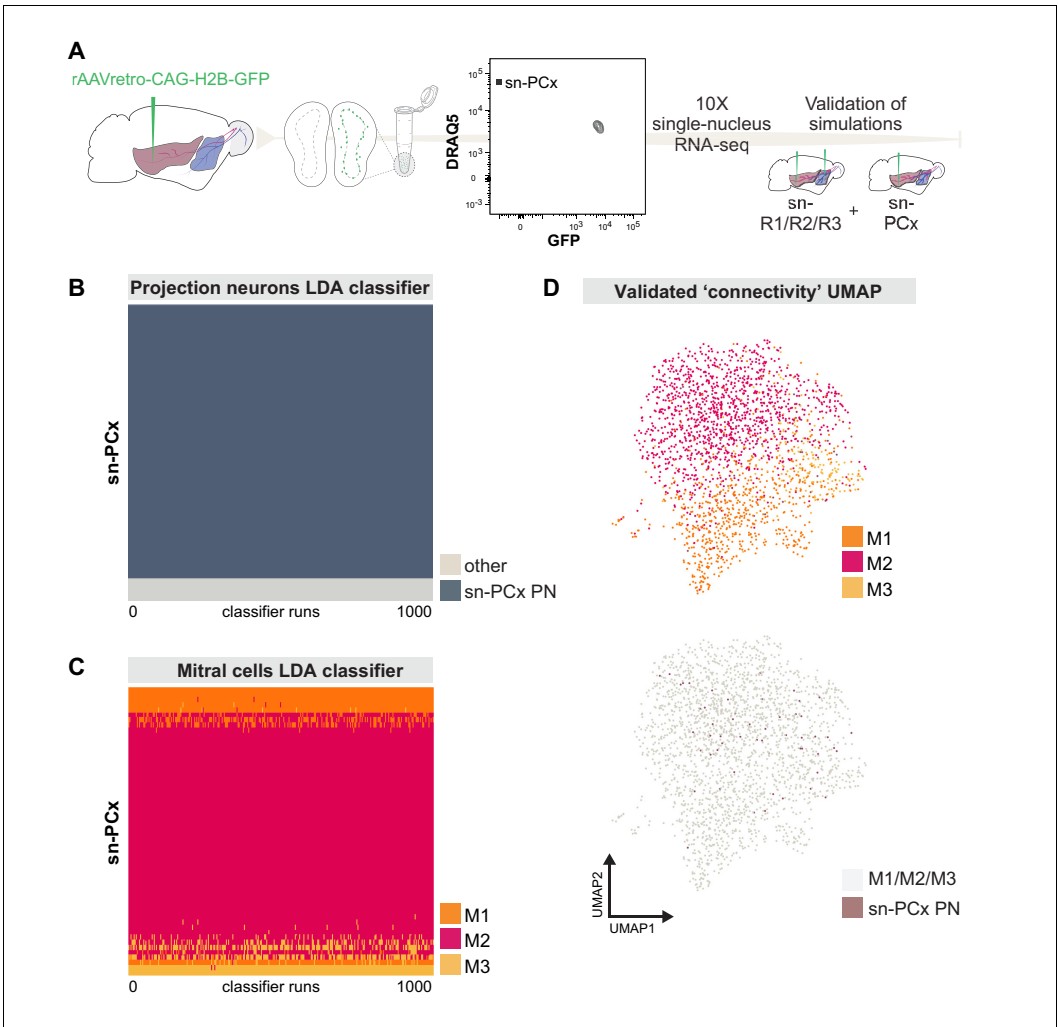

**Figure 7.** Targeted snRNA-seq experiment validates simulation-based predictions of molecularly defined-mitral cell types with distinct projection targets. (**A**) Schematic representation of experimental design. Top: after injection of rAAVretro-CAG-H2B-GFP into PCx, single nuclei were dissociated from one mouse and sorted using Fluorescence-activated Nuclei Sorting (FANS). The population of nuclei is selected based on GFP and DRAQ5 (far-red fluorescent DNA dye). Sorted nuclei were sequenced using 10x single-nucleus RNA-seq and integrated with the main sn-R1/R2/R3 dataset. (**B**) Linear discriminant analysis (LDA) classifiers were trained on the main sn-R1/R2/R3 dataset, then used to predict the projection neuron identity of sn-PCx cells. Each row represents one cell in the new, targeted sn-PCx data. Each column represents one of 1000 LDA classifiers trained to discriminate projection neurons from other cell types. The color indicates the prediction made by each classifier (dark: projection neuron, light: other). (**C**) LDA classifiers were trained on the main dataset sn-R1/R2/R3, then used to predict the cell type identity of sn-PCx projection neurons. Each row represents one sn-PCx projection neuron. Each column represents one of 1000 LDA classifiers trained to predict cell type identity based on regulon expression. Color indicates the predicted cell type identity (orange: M1, red: M2, yellow: M3). (**D**) UMAP showing sn-R1/R2/R3 mitral cells and sn-PCx projection neurons integrated in the same low-dimensional space. UMAP was computed based on regulon expression. sn-PCx projection neurons are in purple, while sn-R1/R2/R3 mitral cells are in color according to their cell type identity (orange: M1, red: M2, yellow: M3) on the top UMAP, while in grey on the bottom (same) UMAP.

Together, these findings validate the predictions made by simulating single-nucleus regulon-based transcriptomes from bulk RNA deep sequencing and suggest that the molecular subcategorization of MCs delineates differences in connectivity (*Figure 7C,D*).

## Discussion

Morphological differences between OB mitral and tufted cells have been described since the time of Cajal (*The croonian lecture, 1894*). Electrophysiological and functional imaging experiments in vivo and in vitro, developmental studies, as well as anatomical reconstructions from light and electron microscopy studies have further highlighted the heterogeneity of OB projection neurons (*Christie et al., 2001*; *Ezeh et al., 1993*; *Fukunaga et al., 2012*; *Geramita et al., 2016*; *Haberly and Price, 1977*; *Kawasawa et al., 2016*; *Mori et al., 1983*; *Phillips et al., 2012*). We here provide the first detailed molecular profiling of mouse OB projection neurons and delineate types and subtypes of mitral and tufted cells together with their key gene regulatory networks and connectivity patterns.

We have performed single-nucleus and bulk RNA deep sequencing to characterize the molecular diversity of mouse OB projection neurons. We identified, based on transcriptome and RNA in situ analysis, three distinct types of MCs and five distinct types of TCs. We then used comprehensive gene regulatory network analysis to reveal potential molecular determinants of cell type-specific functional properties. Finally, we describe a novel computational approach for integrating single nucleus and bulk RNA sequencing data, and we provide evidence that different MC types preferentially project to anterior versus posterior regions of the olfactory cortex. Our analyses provide a comprehensive resource for investigating olfactory circuit function and evolution.

### The molecular diversity of olfactory bulb projection neurons

Given that the vast majority of OB neurons are interneurons, notably granule cells and juxtaglomerular neurons, we devised a retrograde viral targeting strategy to substantially enrich for OB projection neurons. This allowed us to analyze the transcriptomes of over 7500 putative projection neurons that could in turn be grouped into eight molecularly distinct mitral and tufted cell types. We validated neuronal identity using smFISH with multiple type-specific marker genes, and we determined the localization of identified neuronal types within the mitral cell, external plexiform, and glomerular layers of the main olfactory bulb. Finally, we used retrograde viral tracing combined with marker genes to confirm that the molecularly distinct neuronal types we describe indeed project to the olfactory cortex. Based on our analysis, we define three molecularly distinct types of MCs and five distinct types of TCs.

The neuronal cell types we have characterized here likely represent the major categories of OB projection neurons. More extensive sampling might reveal additional rare cell types, and more fine-grained clustering could further subdivide subtypes of neurons. However, our samples contained 7504 putative projection neurons compared to current estimates of 10,000–30,000 projection neurons overall per OB (*Nagayama et al., 2014*; *Richard et al., 2010*). Furthermore, we sorted nuclei rather than whole cells, which is thought to more accurately reflect relative cell-type abundance (*Habib et al., 2017*; *Lake et al., 2017*). Altogether we are therefore confident that our analysis captures the key biologically relevant types of projection neurons. Independent from the number of molecularly distinct neuronal cell types, the gene expression profiles we have described here provide critical new tools for refining projection neuron cell type identities by aligning a cell's molecular features with its functional properties. Previous experiments have highlighted the heterogeneity of odor responses of MCs and TCs (*Balu et al., 2004*; *Bathellier et al., 2008*; *Carey and Wachowiak, 2011*; *Desmaisons et al., 1999*; *Friedman and Strowbridge, 2000*; *Schaefer et al., 2006*). We propose that this functional diversity can be explained, at least in part, by the molecular diversity of OB projection neurons, a model that can now be tested experimentally.

### Specificity of mitral cell projections

A critical feature of neuronal cell type identity is their projection target specificity. Earlier studies have shown that TCs project to anterior regions of the olfactory cortex only, while MCs project to both anterior and posterior olfactory cortex (*Imamura et al., 2020*; *Nagayama et al., 2010*; *Scott et al., 1980*). Furthermore, and in contrast to the organization of neuronal projections to sensory cortexes for vision, hearing, or touch, projections from the OB to the piriform cortex appear to lack apparent topographical organization (*Ghosh et al., 2011*; *Miyamichi et al., 2011*; *Sosulski et al., 2011*, see also *Chen et al., 2021*).

We have analyzed, using bulk RNA deep sequencing, gene expression profiles of cells projecting to anterior versus posterior olfactory cortex (AON versus PCx). Using simulations based on gene regulatory network analysis, we have then mapped the bulk RNA sequencing data onto MC types defined by single-nucleus RNA sequencing. Interestingly, our analysis suggested that cells of the M1 MC type preferentially target the AON, while M2 and M3 cells preferentially target the pPCx. A targeted snRNA-seq experiment of posterior-projecting MCs supported this prediction. Our findings that MC types preferentially target anterior or posterior cortical areas is consistent with recent observations using MAP-seq (*Chen et al., 2021*). Our results are consistent with the model that gene expression and connectivity provide overlapping yet complementary axes for cell-type classification (*Kim et al., 2020*).

## Molecular mechanisms underlying the functional diversity of olfactory bulb projection neurons

Gene regulatory network analysis can reveal the transcriptional programs that determine the functional properties of neuronal subtypes. Here, we describe cell-type-specific modules of gene regulation, defined by the interactions of transcription factors and their target genes. One intriguing result of this analysis is that cell subtypes do not fall into clearly delineated MC and TC classes. For example, module activity of M2b and M2c MC subtypes is more similar to that of T1c TCs than other M1 and M2 MC subtypes. We obtained similar results from analyzing the TF-TF network that is thought to be closely linked to maintenance of cell identity. This demonstrated that while some MC subtypes indeed share key TFs with each other, regulon activity and network features in MC subtypes and TC subtypes are highly overlapping. For example, the most prominent hub, *Pbx3,* is predominantly expressed in M1 and ET2 types. More generally, projection neurons are characterized by a variety of TFs, often closely linked through feedback cycles, that are used by both MC and TC types in a combinatorial manner. The prominent hub and cycle-related genes we have identified here may act as candidate master regulators of neuronal function, which can be targeted for experimental validation. Together, these results suggest that subtypes of MCs and TCs may share important functional properties, possibly blurring at the transcriptional level the traditional division into tufted and mitral cells as the two major classes of OB projection neurons. Moreover, the gradients of module activity that we observed over the MC and TC subtypes theoretically provide a mechanism for generating multiple distinct cellular phenotypes, similar to how morphogenetic gradients allow for spatial patterning and cell differentiation (*Wolpert, 1969*). Through non-linear regulatory interactions, gradual differences at the transcriptomic level can be translated into selective expression of functional genes.

For example, we found that the *Kcng1* gene was selectively expressed in M1 MCs. The *Kcng1* gene encodes for a voltage-gated potassium channel, which forms heterotetrameric channels with the ubiquitously expressed delayed rectifying Kv2.1 potassium channel (indeed also expressed in the M1 cluster) and modifies the kinetics of channel activation and deactivation (*Kramer et al., 1998*). Other voltage-gated potassium channels exhibiting prominent differential expression levels in MC and TC subtypes include *Kcnd3, Kcng1, Kcnh5, Kcnq3, Kcnj2* and 6, and *Hcn1* (for details see accompanying website link in Materials and methods). These channels represent intriguing candidates for controlling the differential excitability of different MC and TC types.

We also found that a large number of cell adhesion and axon guidance genes known to control the formation and maintenance of neuronal connectivity were differentially expressed in OB projection neuron types. Examples include members of the cadherin superfamily of cell adhesion glycoproteins (*Cdh6*, *7*, *8 c*, *9*, *13*, and *20*) and components of the Semaphorin/Neuropilin/Plexin complexes, including *Nrp1, Nrp2, Plexna3, Sema3e,* and *Sema5b*. Semaphorin/Neuropilin/Plexin complexes are known to play critical roles in the development and maintenance of neuronal connections, including in OB MCs (*Inokuchi et al., 2017*; *Saha et al., 2007*). Heterogeneity in these cell adhesion and guidance genes might inform subdivisions in projection neurons across the OB, in particular along the dorsomedial–ventrolateral axis.

Our data set will be an important resource for the emerging study of the evolution of olfactory neural circuits across species. Adaptation to distinct olfactory environments, and the critical role of olfaction in survival and reproduction, has shaped the evolution of the repertoire of odorant receptors and olfactory sensory neurons (*Bargmann, 2006*; *Yoshihito, 2012*). However, little is known about how evolving sensory inputs from the nose are accommodated at the level of the OB and its

connections to the olfactory cortex. The detailed molecular description of mouse OB projection neurons presented here provides an entry point toward understanding the evolution of olfactory sensory processing across species.

# Materials and methods

**Key resources table**

| Reagent type (species) or resource | Designation | Source or reference | Identifiers | Additional information |
|---|---|---|---|---|
| Strain, strain background (*M. musculus*) | C57Bl/6 | Jackson Laboratory | Stock #: 000664 RRID:IMSR_JAX:000664 | |
| Genetic reagent (*Adeno-associated virus*) | rAAVretro-CAG-H2B-GFP | *Tervo et al., 2016* | PMID:27720486 | |
| Commercial assay or kit | Nuclei PURE Prep | Sigma | cat#: NUC201-1KT | |
| Commercial assay or kit | Arcturus PicoPure RNA Isolation Kit | ThermoFisher | cat#: KIT0204 | |
| Other | DRAQ5 | ThermoFisher | cat#: 65-0880-92 | |
| Software, algorithm | Cell Ranger version 3.0 | 10x Genomics | RRID:SCR_017344 | |
| Software, algorithm | Seurat v3.6 | *Butler et al., 2018* | RRID:SCR_007322 | |
| Software, algorithm | Scanpy v1.7.1 | *Wolf et al., 2018* | RRID:SCR_018139 | |
| Software, algorithm | glmGamPoi R-package | *Ahlmann-Eltze and Huber, 2021* | PMID:33295604 | |
| Software, algorithm | SCENIC | *Aibar et al., 2017* | RRID:SCR_017247 | |
| Software, algorithm | Cytoscape | *Shannon et al., 2003* | RRID:SCR_003032 | |
| Software, algorithm | ImageJ version 2.1.0. | *Schindelin et al., 2012* | RRID:SCR_003070 | |

## Experimental model and subject details

Male and female C57Bl/6 mice (6–8 weeks old) were used in this study and obtained by in-house breeding. All animal protocols were approved by the Ethics Committee of the board of the Francis Crick Institute and the United Kingdom Home Office under the Animals (Scientific Procedures) Act 1986 (project license number PA2F6DA12), as well as Brown University's Institutional Animal Care and Use Committee (protocol number: 21-03-0004) followed by the guidelines provided by the National Institutes of Health.

## Stereotaxic injections and histology

Mice were anaesthetized using isoflurane and prepared for aseptic surgery in a stereotaxic frame (David Kopf Instruments). A retrogradely transported Adeno Associated Virus (rAAV-retro-CAG-H2B-GFP, *Tervo et al., 2016*) was injected stereotaxically into multiple sites of piriform cortex (PCx) and anterior olfactory nucleus (AON). The following coordinates, based on the Paxinos and Franklin Mouse Brain Atlas, were used: Coordinates (AP/ML/DV) in mm for PCx injections: (1) −0.63/−4.05/−4.10, (2) −0.8/−4.00/−4.10. For AON injections: (1) 2.8/1.25/2.26 and 2.6, (2) 2.68/1.25/2.3 and 2.75, and (3) 2.34/0.7/3.5. Using a micromanipulator, a pulled glass micropipette was slowly lowered into the brain and left in place for 30 s before the virus was dispensed from the micropipette using a Nanoject injector (Drummond Scientific) at a rate of 46 nl/min (0.3 µl for PCx and 0.2 µl for AON per injection site). The micropipette was left in place for an additional 5 min before being slowly withdrawn to minimize diffusion along the injection tract. Craniotomies were covered with silicone sealant (WPI), and the skin was sutured. Mice were provided with 5 mg/kg Carprofen in their drinking water for 2 days following surgery.

Histology was used to validate viral targeting of olfactory bulb projection neurons. Mice were deeply anaesthetized with 2.5% of 250 mg/kg Avertin and transcardially perfused with 10 ml of ice-cold phosphate-buffered saline (PBS) followed by 10 ml of 4% paraformaldehyde (PFA). Brains were dissected and post-fixed for 5 hr in 4% PFA at 4°C. Coronal sections (100 µm thick) were prepared

using a vibrating-blade Leica VT100S Vibratome. Sections were rinsed in PBS and incubated in PBS/ 0.1% Triton X-100 and Neurotrace counterstain (1:1000, ThermoFisher) at 4°C overnight, then mounted on SuperFrost Premium microscope slides (Fisher, cat# 12-544-7) in Fluorescent Vecta- shield Mounting Medium (Vector). Images were acquired at 20× using a Nikon A1R-HD confocal microscope. While this injection strategy cannot fully exclude labeling of axons of passage (e.g. axons passing through AON), these are unlikely to constitute a significant sub-population of labeled neurons as, e.g., we identified exclusively PCx projecting cell types as well.

## Single-nuclei isolation, FANS, and RNA extraction

To isolate GFP-labeled nuclei, 10 individual biological replicates were used. For bulk RNA deep sequencing: three replicates of AON-injected mice and three replicates of PCx-injected mice (all females, AON1 = 7 w, AON2 = 6w, AON3 = 8w, PCx1 = 7w, PCx2 = 6w, PCx3 = 7w); for single- nucleus RNA sequencing (sn-R1/R2/R3 dataset): three replicates of AON- and PCx-injected mice (all males, 7 w); for validating the simulation results using single-nucleus RNA sequencing (sn-PCx data- set): 1 PCx-injected mouse (female, 11 w). Mice were deeply anaesthetized with an overdose of keta- mine/xylazine and transcardially perfused with ice-cold PBS. Both hemispheres of the olfactory bulb were dissected, and the hemisphere ipsilateral to the injection site was carefully separated from the contralateral hemisphere. Both hemispheres were minced separately and placed into two different tubes. To dissociate single nuclei, Nuclei PURE Prep was used according to the manufacturer instruc- tions (Sigma, cat# NUC201-1KT) with some modifications. The minced tissue was gently homoge- nized in 2.75 ml Nuclei PURE Lysis Buffer and 27.5 µl 10% Triton X-100 using an ice-cold dounce and pestle, and filtered two times through a 40 µm cell strainer on ice. After centrifuging at 500 rpm for 5 min at 4°C, the supernatant was aspirated and gently resuspended in 500 µl of cold buffer (1× of cold Hanks' Balanced Salt Solution HBSS, 1% nuclease-free BSA, 22.5 µl of RNasin Plus [Promega N2611] and 1/2000 DRAQ5). The remaining PBS-perfused brain of the PCx-injected mouse (for sn- PCx dataset) was post-fixed 5 hr in 4% PFA at 4°C and used for histology validation of the injection points.

Fluorescence-activated nuclei sorting of single nuclei was performed using a BD FACSAria III Cell Sorter with a 70 µm nozzle at a sheath pressure of 70 psi. Precision mode (yield mask set to 16, purity mask set to 16 and phase mask set to 0) was used for stringent sorting. For single-nucleus RNA sequencing, GFP+ nuclei were sorted into a 1.5 ml centrifuge tube. For bulk RNA deep sequencing, GFP+ nuclei were sorted into 100 µl Trizol and 1.43 µl of RNA carrier, and total RNA was extracted using the Arcturus PicoPure RNA Isolation Kit (ThermoFisher, cat# KIT0204).

## Single-nucleus RNA sequencing

The sn-R1/R2/R3 cDNA(s) were prepared using the 10× kit version Chromium Single Cell 3' v3. Libraries were prepared using the Next Single Cell/Low Input RNA Library Prep Kit (New England Biolabs). The quality and quantity of the final libraries were assessed with the TapeStation D5000 Assay (Agilent Technologies) before sequencing with an Illumina HiSeq 4000 platform. cDNA con- centrations were measured as: 14.4 (R1), 23.3 (R2), 7.9 (R3) ng/µl (n = 3 animals).

The sn-PCx cDNA and library were prepared using the 10x kit version Chromium Single Cell 3' v3.1. The quality and quantity of the final library was assessed with the TapeStation D1000 Assay (Agilent Technologies) before sequencing with an Illumina NextSeq500. The cDNA concentration was measured as 1.09 ng/µl.

## Bulk RNA deep sequencing

Libraries were prepared using the Next Single Cell/Low Input RNA Library Prep Kit (New England Biolabs). The quality and quantity of the final libraries was assessed with the TapeStation D1000 Assay (Agilent Technologies) before sequencing with an Illumina HiSeq 4000 platform. RNA concen- trations were measured as: AON injections (n = three animals), 1.495, 1.682, 1.881 ng/µl and RNA integrity numbers (RIN) 8.3, 8.7, 9.0; PCx injections (n = 3 animals), 0.257, 0.165, 0.133 ng/µl; RIN = 8.0, 10.0, 7.8 for each replicate, respectively.

## Single-nucleus RNA sequencing analysis

Raw sequencing datasets were processed using Cell Ranger version 3.0 (10x Genomics). Count tables were loaded into R (version 3.6, https://www.r-project.org) and further processed using Seurat v3.6 (*Butler et al., 2018*) and Scanpy v1.7.1 (*Wolf et al., 2018*).

For all four datasets (three biological replicates sn-R1/R2/R3 and sn-PCx), we removed all nuclei with fewer than 1000 distinct genes detected or with more than 5% of unique molecular identifiers stemming from mitochondrial genes. After quality control, for the large sn-RNAseq dataset we merged the three replicates using standard functionality of Seurat: we apply NormalizeData and FindVariableFeatures on the individual replicates, then apply Canonical Correlation Analysis (CCA) (*Stuart et al., 2019*). We retained a total of 31,703 nuclei (median of 2300 genes per nucleus; for each replicate median genes per nucleus: R1 = 2,266; R2 = 2419; R3 = 2,322). For the sn-PCx dataset, we retained a total of 74 nuclei (median genes per nucleus: 6,092). We describe in detail its downstream analysis in the Materials and methods section 'Validation of regulon-based simulations'. For the sn-R1/R2/R3 dataset, principal component analysis (PCA) was then performed on highly variable genes and the first 30 principal components were selected as input for clustering and UMAP, based on manual inspection of a principal component variance plot (PC elbow plot). Clustering was performed using the default method (Louvain) from the Seurat package, with the resolution parameter of the FindClusters function set to 0.3.

Subclustering of projection neurons was carried out by selecting clusters M1, M2/M3, T1, ET1 and ET2 from the initial single-nuclei analysis based on the combinatorial expression patterns of glutamatergic and previously characterized mitral/tufted cell markers (*Tbx21*, *Pcdh21*, *Thy1*, *Vglut1*, *Vglut2*, and *Vglut3*). For subclustered nuclei, we again performed the above steps of highly variable genes selection, principal component analysis, and clustering, this time with the Louvain resolution parameter set to 0.5.

Differential gene expression analysis on single-nuclei data was performed using the glmGamPoi R-package (*Ahlmann-Eltze and Huber, 2021*). Gene set enrichment analysis (GSEA) on the resulting log-fold changes was performed as described in *Subramanian et al., 2005*.

## Network inference

Gene regulatory networks were inferred using the pySCENIC pipeline (Single-Cell rEgulatory Network InferenCe, *Aibar et al., 2017*; *Van de Sande et al., 2020*) and visualized using Jupyter notebooks and Cytoscape (*Shannon et al., 2003*). We used pySCENIC on the projection neuron subcluster data after removal of nuclei assigned the PG cell type. pySCENIC is a three-step approach: (1) predict TF-target gene pairs using Arboreto; (2) filter TF-target gene associations for false positives using TF binding site enrichment in a window of 5 kb around a target's transcription start site (TSS) and group TFs with their target genes into so-called regulons; (3) calculate the activity of regulons in each cell in terms of the area under the recovery curve (AUC). Step one depends on a stochastic search algorithm and is therefore performed n = 100 times. Only TFs that are found >80 times and with TF-target gene interactions that occur >80 times are considered. To avoid technical issues in the analysis, regulons with fewer than eight target genes are removed from the final list. Subsequent analysis in Step three involves a stochastic downsampling to speed up computation; hence, we verified that the chosen sample size was sufficient for accurate AUC approximations. We calculated n = 25 AUC matrices and confirmed that they contained few zeros and the variance of each matrix entry (i.e. approximated regulon activity in a given cell) was low.

## Quantification along cell type trajectories

Gradual and sudden changes between cell types are studied using pseudotime trajectory analysis. We used the partition-based graph abstraction (PAGA) functionality offered by the Python package Scanpy (*Wolf et al., 2019*). Given a set of clusters, PAGA computes a graph as a condensed representation of how the clusters are distributed in transcriptomic/PCA space. It then uses a diffusion map to smoothly travel along the paths in the graph (*Figure 4—figure supplement 3*). We focused on a single path, namely ET4, ET1, T1, M2, M3, M1, ET2 (ignoring ET3, as it has few neurons). We then computed trajectories of module and regulon activity levels, and TF expression levels using a running average with window size of 100 nuclei to smoothen the signal.

## Bulk RNA deep sequencing analysis

The 'Trim Galore!' utility version 0.4.2 was used to remove sequencing adaptors and to quality trim individual reads with the q-parameter set to 20. Sequencing reads were then aligned to the mouse genome and transcriptome (Ensembl GRCm38 release-89) using RSEM version 1.3.0 (*Li and Dewey, 2011*) in conjunction with the STAR aligner version 2.5.2 (*Dobin et al., 2013*). Sequencing quality of individual samples was assessed using FASTQC version 0.11.5 and RNA-SeQC version 1.1.8 (*DeLuca et al., 2012*). Differential gene expression was determined using the R-bioconductor package DESeq2 version 1.24.0 (*Love et al., 2014*). GSEA was conducted as described in *Subramanian et al., 2005*.

## Integration of single-nucleus and bulk RNA deep sequencing data using regulon-based simulations

Nuclei were simulated from bulk RNA deep sequencing data using a weighted random sampling of regulons with replacement. A regulon's relative weight corresponded to the prevalence of its transcription factor in the given bulk RNA-seq sample. Each time a regulon was selected, the counts for its transcription factor and all its target genes increased by one. This weighted random sampling of regulons maintains key information from the starting bulk RNA-seq data (in the form of weights) while also maintaining key co-expression patterns of single nuclei (in the form of regulons), creating a population of biologically-realistic simulated transcriptomes that reflect the starting bulk RNA-seq data. The number of regulons expressed in each simulated nucleus was randomly selected from a list of how many unique transcription factors each nucleus from snRNA-seq expressed (normalized expression > 2). Simulated nuclei were treated as raw count matrices and integrated with snRNA-seq nuclei using the sctransform package in R (*Hafemeister and Satija, 2019*). Briefly, this integration algorithm normalizes counts as the Pearson residuals from a regularized negative binomial regression model with sequencing depth as a covariate. The main result is the removal of batch effects related to sequencing depth, which facilitates pooling of our snRNA-seq replicates and makes the results of the simulation pipeline invariant to the number of regulons chosen for each nucleus, keeping results driven by proportions in the starting bulk RNA-seq data rather than absolute numbers.

To filter simulated nuclei, we trained 1000 LDA classifiers with the python package scikit-learn (*Pedregosa et al., 2011*). For each classifier, snRNA-seq nuclei were split into test and train datasets, with 75% of nuclei used for training and the other 25% used for testing. This train-test split was unique to each classifier. Each classifier was trained to predict whether a nucleus was a projection neuron (i.e. whether it was selected for subclustering in the initial transcriptome-based Seurat analysis) based on values for the 30 top principal components from the sctransform integration. Principal component analysis projects all data points into the same space and reduces the number of features, enabling better classification performance. Each classifier was applied to the remaining snRNA-seq nuclei for testing, and accuracy was evaluated using the Jaccard index calculated by scikit-learn (*Pedregosa et al., 2011*). The classifiers were then applied to the simulated nuclei. Simulated nuclei predicted to be projection neurons by all 1000 classifiers were designated as putative simulated projection neurons and selected for further analysis. Similarly, these putative simulated projection neurons were integrated with snRNA-seq mitral cells using sctransform. One thousand LDA classifiers were trained to classify simulated nuclei as AON-projecting or PCx-projecting based on values for the 30 top principal components from the sctransform integration. Each classifier was trained on 75% of the simulated projection neurons and tested on the other 25%, with accuracy evaluated using the Jaccard index. Each classifier was then applied to snRNA-seq mitral cells.

## Validation of regulon-based simulations

To assign the cell type identity in the sn-PCx dataset, we first identified projection neurons by transforming the expression matrix into the space of 30 principal components defined on the larger sn-R1/R2/R3 dataset. One thousand LDA classifiers were trained to classify cells as projection neurons or not based on the values for these 30 principal components. Each classifier was trained on 75% of the original sn-R1/R2/R3 dataset and tested on the other 25%. sn-PCx cells were considered projection neurons if they were predicted to be such by all 1000 classifiers.

Sn-PCx projection neurons were then transformed into a low-dimensional space defined by their expression of the 86 regulons from our SCENIC analysis on the original sn-R1/R2/R3 projection neurons. These regulons create a low-dimensional space in which to compare this new, targeted sn-PCx dataset to our original, larger sn-R1/R2/R3 dataset. To quantify this comparison, we used LDA classifiers, trained on the original mitral cells, to predict the cell type of the new projection neurons. Each of 1000 LDA classifiers was trained on 75% of the original data and tested on the other 25% to predict cell type identity based on regulon activity scores. These classifiers were then applied to the regulon activity scores of the new projection neurons. Statistics were performed on the results using binomial distributions. To calculate a p-value for the abundance of M2 cells in the sn-PCx dataset, we used the R command pbinom(44,57,1241/2114, lower.tail=F) to quantify the probability of observing that many or more M2 cells in a population of that size were the probability of sampling M2 as opposed to a different MC equal to that of sampling M2 from our sn-R1/R2/R3 dataset. With similar logic, to calculate a p-value for the scarcity of M1 cells in the sn-PCx data, we used the R command pbinom(6,57,711/2114) to calculate the probability of observing that many or fewer M1 cells.

## smFISH in tracing experiments and quantification

Experiments were performed according to the manufacturer's instructions, using the RNAscope Fluorescent Multiplex kit (Advanced Cell Diagnostics [ACD]) for fresh frozen tissue. Briefly, a total of six mice were injected into the AON and PCx with the rAAVretro-CAG-H2B-GFP. After 15 days post-injection, mice were deeply anaesthetized with 2.5% of 250 mg/kg Avertin and transcardially perfused with 10 ml of ice-cold phosphate-buffered saline (PBS). The brains were dissected out from the skull, immediately embedded in Tissue Plus O.C.T. compound (Fisher Healthcare) and snap-frozen in a bath of 2-methylbutane on dry ice. Brains were cryo-sectioned coronally at 20 μm thickness, mounted on Fisherbrand Superfrost Plus microscope slides (Fisher Scientific), and stored at −80 ˚C until use. In situ probes against the following mouse genes were ordered from ACD and multiplexed in the same permutations across sections: *Foxo1* (#485761-C2 and 485761), *Kcng1* (#514181-C2), *Lxh1* (#488581), *Sertm1* (#505401-C2), *Ebf3*(#576871-C3), *Sgcg* (#488051-C3), *Cadps2* (#529361-C3 and 529361), *Coch* (#530911-C3), *Ly6g6e* (#506391-C2), *Wnt5b* (#405051), *Fst* (#454331), *Barhl2* (#492331-C2), *Vdr* (524511-C3), *Gfp* (#409011, #409011-C2 and #409011-C3), *Piezo2* (#500501), *Olfr110/111* (#590641), *Calca* (#578771), *Lhx5* (#885621-C3), and *Vgll2* (#885631-C2). Following smFISH, high-resolution images of a single z-plane were obtained using a 60× oil immersion objective on an Olympus FV3000 confocal microscope and a 40× oil immersion objective on a Nikon A1R-HD confocal microscope.

For each image, we first counted the number of cells positive for the less abundant probe (probe A), on average 60 cells for each probe. We then counted amongst probe A-positive cells how many cells also expressed the second probe (probe B). We report overlap as the percentage of probe A-positive cells also expressing probe B. For mitral cell markers we only considered cells along the mitral cell and internal plexiform layers. For tufted cell markers, we only considered cells in the external plexiform and glomerular layers. We considered a cell positive for a given smFISH probe if more than five dots/cell were present, and if the dots were located within the nucleus and perinuclear compartment of the cell, visualized by DAPI counterstain.

## Acknowledgements

We thank the Crick Advanced Sequencing Facility, especially Robert Goldstone and Amelia Edwards for their excellent support. We thank Debipriya Das and Ana Agua-Doce from the Crick Flow Cytometry Facility for technical assistance, and the Crick animal facility for animal care. We thank the members of the Crick Digital Development Team, particularly Amy Strange, Luke Nightingale, Jude Pinnock, and Marc Pollitt for excellent technical support. We thank the Harvard Neurobiology Imaging Facility for consultation and RNAscope services. This facility is supported in part by the Neural Imaging Center as part of an NINDS P30 Core Center grant #NS072030. We also thank Zach Herbert and Maura Berkeley from the Molecular Biology Core Facilities at the Dana-Farber Cancer Institute for sequencing services. We thank Kevin Carlson of the Flow Cytometry and Sorting Facility at Brown University for his technical assistance. This facility is supported in part by the NCRR equipment grant 1S10RR021051. We thank Keeley Baker, Gilad Barnea, Bob Datta, Kevin Franks, and Paul

Greer for critical comments on the manuscript. Work in the ATS lab was supported by the Francis Crick Institute, which receives its core funding from Cancer Research UK (FC001153), the UK Medical Research Council (FC001153), and the Wellcome Trust (FC001153); a Wellcome Trust Investigator grant to ATS (110174/Z/15/Z), and a DFG postdoctoral fellowship to TA. Work is the AF lab was supported by grants from the NIH (1U19NS112953-01, 1R01DC017437-03) and the Robert J and Nancy D Carney Institute for Brain Science. Carney Institute computational resources used in this work were supported by the NIH Office of the Director grant S10OD025181.

## Additional information

### Competing interests
Andreas T Schaefer: Reviewing editor, *eLife*. The other authors declare that no competing interests exist.

### Funding

| Funder | Grant reference number | Author |
| --- | --- | --- |
| Cancer Research UK | FC001153 | Andreas T Schaefer |
| Wellcome Trust | FC001153 | Andreas T Schaefer |
| Deutsche Forschungsgemeinschaft | AC 304/1-1 | Tobias Ackels |
| National Institutes of Health | 1R01DC017437-03 | Alexander Fleischmann |
| National Institutes of Health | 1U19NS112953-01 | Alexander Fleischmann |
| National Institutes of Health | S10OD025181 | Alexander Fleischmann |
| Medical Research Council | FC001153 | Andreas T Schaefer |
| Wellcome Trust | 110174/Z/15/Z | Andreas T Schaefer |

The funders had no role in study design, data collection and interpretation, or the decision to submit the work for publication.

### Author contributions
Sara Zeppilli, Tobias Ackels, Conceptualization, Formal analysis, Investigation, Visualization, Methodology, Writing - original draft, Writing - review and editing; Robin Attey, Software, Formal analysis, Investigation, Visualization, Methodology, Writing - original draft, Writing - review and editing; Nell Klimpert, Investigation, Visualization, Writing - original draft, Writing - review and editing; Kimberly D Ritola, Writing - review and editing, Essential reagent; Stefan Boeing, Software, Formal analysis, Visualization, Methodology, Writing - review and editing; Anton Crombach, Conceptualization, Software, Formal analysis, Supervision, Investigation, Visualization, Methodology, Writing - original draft, Writing - review and editing; Andreas T Schaefer, Alexander Fleischmann, Conceptualization, Formal analysis, Supervision, Funding acquisition, Investigation, Writing - original draft, Writing - review and editing

### Author ORCIDs
Sara Zeppilli  https://orcid.org/0000-0002-6950-3162
Tobias Ackels  https://orcid.org/0000-0002-4964-1162
Robin Attey  http://orcid.org/0000-0002-9652-8103
Nell Klimpert  https://orcid.org/0000-0002-6166-8026
Kimberly D Ritola  https://orcid.org/0000-0002-5666-2973
Stefan Boeing  http://orcid.org/0000-0003-0495-5659
Anton Crombach  http://orcid.org/0000-0002-2889-5120
Andreas T Schaefer  https://orcid.org/0000-0002-4677-8788
Alexander Fleischmann  https://orcid.org/0000-0001-7956-9096

## Ethics

Animal experimentation: All animal protocols were performed in strict accordance with the recommendations approved by the Ethics Committee of the board of the Francis Crick Institute and the United Kingdom Home Office under the Animals (Scientific Procedures) Act 1986 (project license number PA2F6DA12), as well as Brown University's Institutional Animal Care and Use Committee (protocol number: 21-03-0004) followed by the guidelines provided by the National Institutes of Health.

## Decision letter and Author response

Decision letter https://doi.org/10.7554/eLife.65445.sa1
Author response https://doi.org/10.7554/eLife.65445.sa2

# Additional files

## Supplementary files

• Transparent reporting form

## Data availability

Raw single nucleus RNA sequencing data (large sn-R1/R2/R3 and targeted sn-PCx datasets) have been deposited in Gene Expression Omnibus (GEO) under the accession numbers GSE162654 and GSM5363097, respectively. Bulk RNA deep sequencing data has been deposited in GEO under the accession number GSE162655. The R and Python analysis scripts developed for this paper are available at the GitLab links https://gitlab.com/fleischmann-lab/papers/molecular-characterization-of-projection-neuron-subtypes-in-the-mouse-olfactory-bulb (copy archived at https://archive.softwareheritage.org/swh:1:rev:4bbee4380a84756ea0e6532f5603b2d99c8e62fa) and https://gitlab.inria.fr/acrombac/projection-neurons-mouse-olfactory-bulb (copy archived at https://archive.softwareheritage.org/swh:1:rev:b9d253e7885124d810b52a0cd511bf8032b5efa5). Extensive computational tools for additional in-depth exploration of our data sets are available through our website: https://biologic.crick.ac.uk/OB_projection_neurons.

The following datasets were generated:

| Author(s) | Year | Dataset title | Dataset URL | Database and Identifier |
|---|---|---|---|---|
| Zeppilli S, Ackels T, Attey R, Klimpert N, Ritola KD, Boeing S, Crombach A, Schaefer AT, Fleischmann A | 2020 | single nucleus RNA sequencing data | https://www.ncbi.nlm.nih.gov/geo/query/acc.cgi?acc=GSE162654 | NCBI Gene Expression Omnibus, GSE162654 |
| Zeppilli S, Ackels T, Attey R, Klimpert N, Ritola KD, Boeing S, Crombach A, Schaefer AT, Fleischmann A | 2020 | RNA deep sequencing data | https://www.ncbi.nlm.nih.gov/geo/query/acc.cgi?acc=GSE162655 | NCBI Gene Expression Omnibus, GSE162655 |
| Zeppilli S, Ackels T, Attey R, Klimpert N, Ritola KD, Boeing S, Crombach A, Schaefer AT, Fleischmann A | 2021 | single nucleus RNA sequencing data | https://www.ncbi.nlm.nih.gov/geo/query/acc.cgi?acc=GSM5363097 | NCBI Gene Expression Omnibus, GSM5363097 |

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
