## [Decision Letter]

**Acceptance summary:**

The authors performed single cell RNA sequencing in combination with retrograde labeling from the anterior olfactory nucleus and the anterior piriform cortex, to reveal several distinct cell types of projection neurons in the olfactory bulb. The authors further characterized gene regulatory networks and the relationship between gene expressions and their axonal projections. This study provides foundational information and resource regarding the diversity of projection neurons in the olfactory bulb.

**Decision letter after peer review:**

Thank you for submitting your article "Molecular characterization of projection neuron subtypes in the mouse olfactory bulb" for consideration by *eLife*. Your article has been reviewed by 3 peer reviewers, and the evaluation has been overseen by a Reviewing Editor and Catherine Dulac as the Senior Editor. The reviewers have opted to remain anonymous.

The reviewers have discussed the reviews with one another and the Reviewing Editor has drafted this decision to help you prepare a revised submission.

We would like to draw your attention to changes in our policy on revisions we have made in response to COVID-19 (https://elifesciences.org/articles/57162). Specifically, when editors judge that a submitted work as a whole belongs in *eLife* but that some conclusions require a modest amount of additional new data, as they do with your paper, we are asking that the manuscript be revised to either limit claims to those supported by data in hand, or to explicitly state that the relevant conclusions require additional supporting data.

Essential revisions:

Summary:

This study by Zeppilli et al. addresses the question whether the projection neurons in the olfactory bulb (OB) are different in their molecular identity. This is one of the major open questions in the olfaction field. The authors enriched the projection neuron population through retrograde tracing from the anterior olfactory nucleus (AON) and piriform cortex (PCX), and performed single nucleus RNA-Seq. They identified 3 mitral cell clusters and 5 tuft cell clusters. They further leveraged regulon analysis (Single-Cell Regulartory Network Inference and Clustering, SCENIC) and found more subtypes based on the transcription factor activities. They developed a simulation method and used bulk RNA-seq data traced from AON and PCX separately to infer that projection neurons of different molecular identity project differently to the targets (however, see Essential revision #1 below).

For the most part our understanding of these projection neurons from a molecular perspective rounds to nearly zero, and their diversity is likely critical for region-specific functions of the higher olfactory brain. The data will be a useful resource to the community, and we commend the authors on releasing the data along with interactive tools for exploring it. In addition, this paper appears to be a useful proof-of-concept for the use of single cell data to deconvolute bulk sequencing data.

Overall, all the reviewers thought that the data are presented clearly and the results are important. However, the reviewers raised several important concerns to which we hope the authors can address with additional experiments and analyses. Most importantly, the reviewers were concerned that the projection specificity predicted with simulated cells (Figure 6) is not validated with real data although projection patterns are one of the major subjects of the study. We hope that the authors can validate the results with additional experiments.

1) The AAVretro labeling strategy is supposed to enrich sequencing of OB projection neurons, which account for only a few percent of OB neurons. However, for unknown reason, only <24% of single nuclei are from projection neurons (mitral and tufted cells). The rest are all kinds including non-neuronal cells. While this does not affect the single-nucleus RNA-Seq analysis, as the authors can identify projection neurons based on known markers and only focus on those 24% cells, they also used the same AAVretro method to perform bulk RNA-Seq, which presumably has the same contamination issues. And in bulk RNA-Seq, RNAs from the other three-quarter cells cannot be distinguished from RNAs from projection neurons. No matter how sophisticated analysis they are doing, I just don't see how the authors could conclude projection patterns from two different retrograde AAV experiments given the above contamination. To make matters worse, the contamination is most likely due to AAV spread, and therefore the AON injection would have more contaminations of OB cells than PCX contaminations so these two bulk RNA-Seq samples contain (unknown) different amount of OB cells. To validate their conclusions of Figure 6, the authors need to either perform classical retrograde labeling together with RNAscope detection of cell-type-specific marker, or better yet to perform AAVretro-based single-nucleus RNA-Seq using separate AAVretro injection (perhaps a smaller volume) into AON and PCX (not sure why they injected two sites in the same animal for their main experiment). Alternatively, the authors could potentially use cell-type specific genetic labeling approaches using novel markers identified in the profiling and mapping target fields. As projection patterns are a framing question for this study, we would like to see at least one experimental verification on this.

2) The computational approach for sorting the contributions of projection neuron populations to the bulk sequencing data is novel and interesting, but not entirely convincing or well-validated. In addition to the issues discussed above, the extent to which simulated nuclei capture the diversity of the original single nucleus RNA-Seq data remains ambiguous since the simulated and actual data seem to form separate clusters in the shared UMAP space. It would be useful to see a simpler complimentary analysis of the bulk RNA-Seq data, perhaps via approaches that evaluate how much of the difference between the AON-projecting and PCX-projecting could be explained by over/under representation of certain clusters in each dataset.

3) In comparing the SCENIC data and analysis shown herein with the Seurat based analysis, there seem to be differences both at the gene level and at the cell type level. Can the authors more extensively characterize similarities and differences in these orthogonal modes of analysis? What explains the observed differences (for example, several of the marker genes shown in Figure 3 are not identified in regulons or TF networks in Figures 4 and 5)?

4) Given that SCENIC is a bioinformatic method, all of the conclusions about regulatory relationships are inferred; the authors should be careful not to assert mechanistic causality from these sorts of analyses (e.g., "continuous activity gradient of TFs is transformed in a non-linear manner into distinct transcriptome differences between mitral and tufted cell types"), which will ultimately require additional experiment. Some tempering of language is likely called for here.

5) A previous single cell RNA-Seq study (Tepe et al., 2018, Cell Report) from OB has identified 3 projection neuron types; those authors' approach resulted in significant labeling of non-projection neurons, about 78% of the data. Since the current method also does not strictly label projection neurons, the authors identified projection neurons from the data using known molecular markers. Therefore, the current method only offered moderate advantage over the previous study. How are the projection neuron cell types correlate with the previously identified 3 types? We note that the quality of the single-cell transcriptomes appears to be superior than the Tepe et al. based on genes detected per cell. Incidentally, it will be useful to plot parameters of snRNA-seq as supplement figure panels rather than bury them in Methods.

6) What is the rationale of using Seurat 0.3 resolution rather than higher or lower? Did the authors try higher or lower resolutions? What would happen to cell-type-specific marker expression? To the regulon analysis? It will be more reassuring, if variations of cluster resolution does not change the major conclusions.

7) It is unclear how the transcription factors identified in the regulon analysis specify these cell types. Their expression patterns do not seem to be cell type specific. Some of them are generic transcription factors and regulate functions in other cells. Examples are Sox10, Sp1, Fos, Jun, Egr1. The authors should examine the hub transcription factors using in situ or at least plot their expressions in their own dataset to make sure they do express in the projection neurons. In addition, the previously identified transcription factors, such as Tbr1, Tbr2 and Tbx21 genes are not in the analysis. The authors should discuss the relationship of the transcription factors identified here and the previously known projection neuron transcription factors.

8) The sub clustering of projection neurons from single nucleus RNA-Seq data is elegant and well-validated with convincing fluorescent in situ hybridization (FISH) data. However, some sort of quantification of overall labeling and overlap between different FISH probes would be useful. Related, it would be broadly beneficial to the field to provide such quantification, particularly for choosing cell-type specific markers in the future.

9) Although compelling, the authors may be somewhat stretching the extent to which the gene regulatory network (GRN) clustering recapitulates the transcriptomic clustering. To strengthen this, it may be useful to include some sort of quality measure for GRN clustering to evaluate if class changes are truly meaningful. This might be achieved by clustering based only on the GRN analysis which is not constrained by the originally identified cell classes. This seems important given that the authors use the rationale that there is heterogeneity of active GRNs within transcriptomically identified clusters to justify the sub clustering, and point out that gradual transitions may exist in active GRNs between clusters. This seems to be a fundamental distinction – whether the GRNs functionally define cell types or whether they are a separate axis along which to evaluate cell identity.

10) The methods and description of the simulated nuclei approach are somewhat under-described. There are multiple manipulations to the bulk sequencing gene expression (expanding the bulk sequencing data into many simulated nuclei by resampling the counts for regulon/target genes, transforming the simulated counts into Pearson residuals before performing PCA and multiple rounds of classification), but the rationale for and the effects of each of these transformations are relatively unclear. Comparisons to existing algorithms that can deconvolve bulk RNA sequencing data (e.g. Frishberg 2019 Nat Methods, Newman 2019 Nat Biotechnology), would also be useful.

[Editors' note: further revisions were suggested prior to acceptance, as described below.]

Thank you for submitting your article "Molecular characterization of projection neuron subtypes in the mouse olfactory bulb" for consideration by *eLife*. Your article has been reviewed by 3 peer reviewers, and the evaluation has been overseen by a Reviewing Editor and Catherine Dulac as the Senior Editor. The reviewers have opted to remain anonymous.

Essential Revisions:

All the reviewers commend the authors' thorough and thoughtful revisions. There are, however, some points that we would like you to address before publication of this work. You can find these points in the individual reviewers' comments. We expect that modifications of text will be sufficient to address these points. We hope that you can address these points within two weeks.

*Reviewer #1 (Recommendations for the authors):*

The authors have addressed many critiques of the reviewers quite thoroughly. Furthermore, they have performed new single-nucleus RNA-sequencing of mitral cells based on their projections into specific sites. The data provide some support of their previous predictions that specific mitral types have preference for specific target sites.

While I applaud the authors' effort to perform additional single nucleus RNAseq of mitral cells based on their projection to either PCx or AON, the numbers of recovered cells are quite small-they did not report any data on AON-projecting ones because the cell number is too small. Of the 57 cells from the sn-PCx data as projection neurons 44 were classified as M2 cells while 6 were classified as M1 cells. Thus, the heading "Targeted snRNA-seq validates predictions of selective connectivity for molecularly defined mitral cell types" is an overstatement: either the prediction is not completely accurate, or the selectivity is not absolute, or both. This heading and similar statements throughout the text need to be toned down.

*Reviewer #2 (Recommendations for the authors):*

This a thoughtful response to all previous reviewers with extensive new data analyses, figure modifications, and clarifications to the text. This is a vastly improved manuscript, and a timely contribution to the field. I commend the authors on a nice revision and fully endorse publication at *eLife*.

*Reviewer #3 (Recommendations for the authors):*

I commend the authors for performing a thorough revision of the paper that addressed our main concerns, especially for the validation data shown in Figure 7 and for the additional details regarding both clustering and the imputation of single nucleus transcriptomes. This paper will serve as an important catalog of molecular diversity in the bulb. I have one experimental request and two minor comments. The experimental request was made in our initial review but was not addressed – unless I missed it (and I may indeed have done so), the authors did not validate the expression of their three key hub genes in MT cells (Taf1, Bclaf1, Pbx3) by either in situ or by showing a UMAP plot describing their expression in the relevant cell types – this is really critical for believing the analysis shown in Figure 5 and should be provided. The two minor comments are 1. maybe consider not using the word "direct" in the abstract to describe OSN inputs onto projection neurons, given the complexity and 2. Line 1276 should define "PCC" – it took me a second to understand what that was on like 1277.

---

## [Author Response]

Essential revisions:Summary:This study by Zeppilli et al. addresses the question whether the projection neurons in the olfactory bulb (OB) are different in their molecular identity. This is one of the major open questions in the olfaction field. The authors enriched the projection neuron population through retrograde tracing from the anterior olfactory nucleus (AON) and piriform cortex (PCX), and performed single nucleus RNA-Seq. They identified 3 mitral cell clusters and 5 tuft cell clusters. They further leveraged regulon analysis (Single-Cell Regulartory Network Inference and Clustering, SCENIC) and found more subtypes based on the transcription factor activities. They developed a simulation method and used bulk RNA-seq data traced from AON and PCX separately to infer that projection neurons of different molecular identity project differently to the targets (however, see Essential revision #1 below).For the most part our understanding of these projection neurons from a molecular perspective rounds to nearly zero, and their diversity is likely critical for region-specific functions of the higher olfactory brain. The data will be a useful resource to the community, and we commend the authors on releasing the data along with interactive tools for exploring it. In addition, this paper appears to be a useful proof-of-concept for the use of single cell data to deconvolute bulk sequencing data.Overall, all the reviewers thought that the data are presented clearly and the results are important. However, the reviewers raised several important concerns to which we hope the authors can address with additional experiments and analyses. Most importantly, the reviewers were concerned that the projection specificity predicted with simulated cells (Figure 6) is not validated with real data although projection patterns are one of the major subjects of the study. We hope that the authors can validate the results with additional experiments.

Thank you very much for the support and the important and helpful suggestions, criticism and comments. As a consequence, we have re-done both the Seurat and pySCENIC analyses and regenerated the relevant parts of manuscript Figures 2-6. This also allowed us to iron out a few minor inconsistencies in the analysis, which have altered some results in a minor way but have not changed any of the conclusions. Regarding the small changes in results, these are essentially restricted to the pySCENIC analysis (Figures 4 and 5). One should think of, for example, some regulons no longer meeting the 0.8 confidence cutoff, whilst others now do. In other words, some regulons will be found missing and some newcomers are seen.

1) The AAVretro labeling strategy is supposed to enrich sequencing of OB projection neurons, which account for only a few percent of OB neurons. However, for unknown reason, only <24% of single nuclei are from projection neurons (mitral and tufted cells). The rest are all kinds including non-neuronal cells. While this does not affect the single-nucleus RNA-Seq analysis, as the authors can identify projection neurons based on known markers and only focus on those 24% cells, they also used the same AAVretro method to perform bulk RNA-Seq, which presumably has the same contamination issues. And in bulk RNA-Seq, RNAs from the other three-quarter cells cannot be distinguished from RNAs from projection neurons. No matter how sophisticated analysis they are doing, I just don't see how the authors could conclude projection patterns from two different retrograde AAV experiments given the above contamination. To make matters worse, the contamination is most likely due to AAV spread, and therefore the AON injection would have more contaminations of OB cells than PCX contaminations so these two bulk RNA-Seq samples contain (unknown) different amount of OB cells. To validate their conclusions of Figure 6, the authors need to either perform classical retrograde labeling together with RNAscope detection of cell-type-specific marker, or better yet to perform AAVretro-based single-nucleus RNA-Seq using separate AAVretro injection (perhaps a smaller volume) into AON and PCX (not sure why they injected two sites in the same animal for their main experiment). Alternatively, the authors could potentially use cell-type specific genetic labeling approaches using novel markers identified in the profiling and mapping target fields. As projection patterns are a framing question for this study, we would like to see at least one experimental verification on this.

Thank you for this important suggestion. As suggested by the reviewers, we have now performed small volume AAV-retro injections, separately into the AON and the pPCx (Author response image 1). Analysis of these new experiments validate our predictions, based on regulon-based simulations, that mitral cell subtypes project to distinct target areas.

In detail, we injected a retrogradely transported Adeno-Associated Virus expressing nuclear GFP (rAAV-retro-CAG-H2B-GFP (Tervo et al., 2016)) into the AON (n=1 animal) or in the pPCx (n=1 animal). Injection coordinates were the same as for the main experiment, we only reduced the volume of each injection site and eliminated the most lateral injection point of AON coordinates to avoid as much LOT contamination as possible. We dissected the olfactory bulbs of the two AON- and pPCx-injected mice, the bulbs were dissociated into single nuclei and enriched for GFP expression using FANS (ipsilateral bulb for enrichment, contralateral bulb as control). We then performed snRNA-seq using 10X Genomics technology. The remaining part of the brain was postfixed in 4 % PFA at 4 °C and used for histological validation of the injection points.

We have now included this new data as Figure 7 and describe the results in detail in the new revised Results section (page 17, line 424-454). We have also made these new datasets available through GEO under the accession number GSE176350.

**Author response image 1. sa2fig1:** Schematic representation of experimental design and Coronal section. (A) Schematic representation of experimental design. Top: after injection of rAAVretro-CAGH2B-GFP into PCx, single nuclei were dissociated from 1 mouse and sorted using Fluorescence-activated Nuclei Sorting (FANS). The population of nuclei is selected based on GFP and DRAQ5 (far-red fluorescent DNA dye). Sorted nuclei were sequenced using 10X single-nucleus RNA-seq and integrated with the main dataset sn-R1/R2/R3. Bottom: Same as the top figure but injection was performed into AON. (B) Coronal section showing GFP expression in the injection points for PCx (top) and AON (bottom) after injection of rAAVretro-CAG-H2B-GFP. Neurotrace counterstain in blue. Scale 50μm. PCx: Piriform Cortex; AON: Anterior Olfactory Nucleus.

2) The computational approach for sorting the contributions of projection neuron populations to the bulk sequencing data is novel and interesting, but not entirely convincing or well-validated. In addition to the issues discussed above, the extent to which simulated nuclei capture the diversity of the original single nucleus RNA-Seq data remains ambiguous since the simulated and actual data seem to form separate clusters in the shared UMAP space. It would be useful to see a simpler complimentary analysis of the bulk RNA-Seq data, perhaps via approaches that evaluate how much of the difference between the AON-projecting and PCX-projecting could be explained by over/under representation of certain clusters in each dataset.

Thank you very much for this comment. As discussed above, we now have independently validated our computational approach with new, targeted snRNA-seq experiments. The distribution of cell types observed in this new data corroborates the predictions made according to the simulation pipeline. We have considered alternate computational approaches (see also our reply to point 10 below), however, we are not convinced that these would resolve the ambiguities mentioned above, considering that most other methods (such as Zhu et al., Sci Rep 6:38350, Cobos et al. Bioinformatics, 34:1969) tend to take less information into account. We are, however, convinced that the new experimental data provides a significantly stronger validation.

To assign the cell type identity of new snRNA-seq cells, we first identified projection neurons by transforming the expression matrix into the space of 30 principal components defined on the original, larger snRNA-seq dataset. 1000 Linear discriminant analysis (LDA) classifiers (Author response image 2), each trained on 75% of the original snRNA-seq dataset and validated on the other 25% (mean jaccard index = 0.96) designated 57 out of 62 cells from the sn-PCx dataset as projection neurons and 14 out of 118 cells from the sn-AON dataset (consistent with the sn-R1/R2/R3 data presented originally that contained “contaminations” from non-projection neurons as well). Given the paucity of sn-AON projection neurons, we chose to focus on only sn-PCx neurons for further analysis.

**Author response image 2. sa2fig2:** Each row represents one cell in the new, targeted snRNA-seq data. Each column represents one of 1000 linear discriminant analysis (LDA) classifiers trained to discriminate projection neurons from other cell types (analogously to the analysis in the original manuscript Figure 7B). The color indicates the prediction made by each classifier (dark: projection neuron, light: rest).

To assign cell type to these projection neurons, these projection neurons were then transformed into a low-dimensional space defined by their expression of the 86 regulons from our SCENIC analysis on the original sn-R1/R2/R3 projection neurons. These regulons create a lowdimensional space in which to compare this new, targeted sn-PCx dataset to our original, larger sn-R1/R2/R3 dataset. To quantify this comparison, we used LDA classifiers, trained on the original mitral cells, to predict the cell type of these new projection neurons. Each of 1000 LDA classifiers was trained on 75% of the original data and validated on the other 25% to predict cell type identity based on regulon activity scores. These classifiers were then applied to the regulon activity scores of the new projection neurons (Author response image 3). Out of 57 sn-PCx mitral cells, only 6 were classified as M1 by at least 80% of classifiers. This scarcity of M1 neurons would be very unlikely if M1 cells were present in the sn-PCx mitral population at the same rate as the general mitral population observed in our original sn-R1/R2/R3 sample, as modeled with a binomial distribution (p=5.57e-05). Therefore, the new data suggest that M1 mitral cells preferentially project to the AON, as predicted by the analyses using simulated transcriptomes. Additionally, 44 of 57 sn-PCx mitral cells were classified as M2 cells by at least 80% of classifiers. Were M2 cells present in the sn-PCx population at the same rate as in the general mitral population, observing this many or more M2 cells would be unlikely, as modeled with a binomial distribution (p=0.001). Therefore, the new data suggest that M2 mitral cells preferentially project to the PCx, as predicted by the analyses using simulated transcriptomes.

We now include this new data, analysis and description as Figure 7 in the revised manuscript. Please also see response to point (10) below.

**Author response image 3. sa2fig3:** Linear discriminant analysis (LDA) classifiers were trained on the original, larger snRNA-seq dataset, then used to predict the cell type identity of projection neurons in a new, targeted snRNA-seq dataset. Each row represents one projection neuron in the new, targeted sn-PCx data. Each column represents one of 1000 linear discriminant analysis (LDA) classifiers trained to predict cell type identity based on regulon expression. Color indicates the predicted cell type identity (orange: M1, red: M2, yellow: M3).

3) In comparing the SCENIC data and analysis shown herein with the Seurat based analysis, there seem to be differences both at the gene level and at the cell type level. Can the authors more extensively characterize similarities and differences in these orthogonal modes of analysis? What explains the observed differences (for example, several of the marker genes shown in Figure 3 are not identified in regulons or TF networks in Figures 4 and 5)?

As suggested by the reviewers, we have performed a more comprehensive comparison between the SCENIC and Seurat-based analyses.

First, we revised the mapping of Seurat’s clusters on SCENIC’s and vice versa . We now compare in a straightforward manner Louvain clustering in Seurat’s PCAspace against Louvain clustering in SCENIC’s regulon activity space. In summary, even if the two analysis modes are orthogonal, neurons considered similar according to one method remain close to each other (i.e., are considered similar) also in the other method.

The comparison resulted in a new Figure 4 —figure supplement 1 in the revised manuscript. One can appreciate that the mapping of regulon-based clusters on Seurat’s UMAP shows both consistent clusters (ET4), grouped clusters (M1 and M3) and gradient-like overlapping clusters (mostly ET1, T1, M2).

While overall comparable, the mapping between Seurat and SCENIC clusters also reveals some differences (Author response image 4; Adjusted Rand Index (ARI) = 0.326, in the revised manuscript as the new Figure 4 —figure supplement 1C). When we do the reverse exercise of mapping Seurat’s clusters onto the UMAP computed for pySCENIC, we find that each of Seurat’s clusters occupies a distinct region of the UMAP. This means clustering results have remained consistent, even if the Louvain algorithm in SCENIC-space partitions the data differently (in the revised manuscript as the new Figure 4 —figure supplement 1D-F).

Second, we now studied how marker genes identified in the Seurat analysis manifest in the SCENIC analysis (see also Comment 7). We have taken marker genes from manuscript Figure 3 and checked at which confidence level they were present as regulons and/or as target genes. The tables below provide a simplified overview, we have incorporated a regulon module-oriented version in Figure 4 —figure supplement 2 of the revised manuscript. In addition, we now provide a supplementary figure to Figure 5A (Figure 5 —figure supplement 1) to show what the network looks like with marker genes added and highlighted by confidence level and highlight the new Figure 5—figure supplement 2 describing differentially active TF networks on p. 14 (l. 356).

**Author response image 4. sa2fig4:** Seurat marker genes identified as regulons in pySCENIC. Columns are cell type, marker gene, and the marker gene as regulon with its max confidence cutoff. Any regulon with a confidence cutoff <0.3 is ignored.

**Author response image 5. sa2fig5:** Presence of marker genes identified through Seurat as target genes in regulons of SCENIC. Columns from left to right are: cell type; marker gene; presence in regulon(s) with corresponding confidence cutoffs. Any regulon with a confidence cutoff <0.4 is ignored.

The third and final aspect of our comparison between Seurat and SCENIC refers to the regulon gradients that we loosely identified in the original version of the manuscript (original Figure 4F).

These can now be interpreted as the mismatch in the two analysis approaches. We refer to Comment 32 for a more in-depth treatment of this topic.

The full comparison of the two orthogonal methods, including the new analysis, is part of the new Figure 4, Figure 4 —figure supplements 1 and 3 and discussed on page 11 of the revised manuscript.

4) Given that SCENIC is a bioinformatic method, all of the conclusions about regulatory relationships are inferred; the authors should be careful not to assert mechanistic causality from these sorts of analyses (e.g., "continuous activity gradient of TFs is transformed in a non-linear manner into distinct transcriptome differences between mitral and tufted cell types"), which will ultimately require additional experiment. Some tempering of language is likely called for here.

The reviewers raise an important point here and we are glad to be given the opportunity to avoid such confusion. Specifically, they refer to a phrase in the Results section that is better placed in the Discussion section. Indeed, we now discuss gradients on p. 22, l. 550, where we offer a potential explanation of how a gradient of module activity (average activity of group of TFs and their target genes) may result in distinct phenotypic outcomes, namely distinct mitral and tufted cell types. We did not intend to assert mechanistic causality from our bioinformatic analysis, but to offer a hypothesis that can direct future experiments.

In addition, we kindly refer to comment 32 for an in-depth revision of the quantification of activity gradients.

5) A previous single cell RNA-Seq study (Tepe et al., 2018, Cell Report) from OB has identified 3 projection neuron types; those authors' approach resulted in significant labeling of non-projection neurons, about 78% of the data. Since the current method also does not strictly label projection neurons, the authors identified projection neurons from the data using known molecular markers. Therefore, the current method only offered moderate advantage over the previous study. How are the projection neuron cell types correlate with the previously identified 3 types? We note that the quality of the single-cell transcriptomes appears to be superior than the Tepe et al. based on genes detected per cell. Incidentally, it will be useful to plot parameters of snRNA-seq as supplement figure panels rather than bury them in Methods.

Tepe et al. identified 3 clusters of putative mitral and tufted cell projection neurons, named M/TC1, 2, and 3. We have performed an integration between the two data sets and highlighted in colors our sub clustering group. While difficult to directly compare single cell and single nucleus RNAseq datasets, it is apparent from the integration and QC plots that we were able to enrich significantly for projection neurons and therefore able to identify different subtypes with a finer-grained resolution (Author response image 6 and Author response image 7).

**Author response image 6. sa2fig6:** Integration of nuclei from R1/R2/R3 snRNA-seq with cells from Tepe et al. 2018 dataset.(A,B) Integration of WT replicates from the Tepe et al. study with the integrated R1/R2/R3 snRNAseq dataset of the present study. Count data for samples WT1 and WT2 were extracted from file GSE121891_OB_6_runs.raw.dge.csv.gz. All WT1 and WT2 cells listed in the GSE121891_OB_metaData_seurat.csv.gz metadata file were used for the integration with the QCfiltered nuclei from the R1/R2/R3 snRNAseq dataset. Data integration between the two full datasets and label transfer of only projection neuron clusters (Figure 2D in the manuscript) to both full datasets were performed using the Seurat 3 R-package. MC/TC clusters are color-coded as indicated.(C,D) Quantification of cells/nuclei labelled as MC/TC clusters in the two datasets.

**Author response image 7. sa2fig7:** On the left is shown the number of genes per cell in the Tepe et al. , 2018 dataset while on the right the equivalent QC parameter (gene/nucleus) in our R1/R2/R3 snRNAseq dataset.

6) What is the rationale of using Seurat 0.3 resolution rather than higher or lower? Did the authors try higher or lower resolutions? What would happen to cell-type-specific marker expression? To the regulon analysis? It will be more reassuring, if variations of cluster resolution does not change the major conclusions.

**Author response image 8. sa2fig8:** UMAPs for different numbers of neighbors (nb, rows) and different Louvain resolutions (res, columns). Neighbors are taken at 25, 50 and 100; resolutions at 0.2, 0.3, 0.5, and 0.7. We recomputed the PCA space of the projection neuron subcluster and consequently the displayed clustering is slightly different from the one in the original manuscript (See also comment 38). ‘nc’ stands for number of detected clusters, ‘ari’ is the adjusted rand index in comparison to the clustering of the original manuscript.

Thanks for pointing out this source of confusion. We have now repeated the clustering for a variety of different parameters (different Seurat resolutions as well as number of neighbors; Author response image 8). Our original clustering parameters were tuned to get as close as possible to biologically-relevant cell types. We initially looked at cross-correlation matrices for distinct resolution parameters, that were computed on the top highly differentially expressed genes between the clusters. We then looked for specific/unique marker genes for each cluster to make sure that at least a few marker genes could differentiate one cluster from another. Finally, we selected the specific marker genes for in situ validation that corroborated our first-level clustering. We are confident that here we are describing the main classification of mitral and tufted cells, in line with the projection-neuronconnectivity axis as well. Our main groups could be further split into subtypes based on additional biological axes taken into consideration, nonetheless our classification will serve as a general roadmap to better investigate functional differences between main subclasses projecting to different areas of the brain.

Finally, cell types as defined by Seurat clustering influence regulon analysis in the sense that we decided to remove the PG cluster for the pySCENIC analysis (i.e., we wanted to only study M/T neurons). As the PG cluster is very stable across neighborhood and resolution changes (see central cluster in UMAPs of Author response image 8) this influence is “constant” and does not affect the major conclusions of our study. We now highlight the resolution parameters on p11 as well as in the methods section (p 27) of the revised manuscript. We also now clarify our use of the term “type” and “subtype” in the revised manuscript in Figure 4.

7) It is unclear how the transcription factors identified in the regulon analysis specify these cell types. Their expression patterns do not seem to be cell type specific. Some of them are generic transcription factors and regulate functions in other cells. Examples are Sox10, Sp1, Fos, Jun, Egr1. The authors should examine the hub transcription factors using in situ or at least plot their expressions in their own dataset to make sure they do express in the projection neurons. In addition, the previously identified transcription factors, such as Tbr1, Tbr2 and Tbx21 genes are not in the analysis. The authors should discuss the relationship of the transcription factors identified here and the previously known projection neuron transcription factors.

We apologize for not making the rationale behind the regulon analysis clearer in the original manuscript. Regulon analysis does not solely produce a list of cell-type specific transcription factors. It will find both general and specific factors. Indeed, precisely to distinguish between general and specific factors in the context of M/T cell types, we highlighted in Figure 5 cell-type specific regulons. To make our analysis more transparent, we now add a supplementary figure to Figure 5 (Figure 5 figure supplement 1A in the revised manuscript) that shows which regulons are found to be specific for each cell type.

**Author response image 9. sa2fig9:** UMAPs of the reviewer’s requested TFs: Sox10, Sp1, Fos, Jun, Egr1. All TFs, with the exception of Sox10, are expressed in the projection neurons. We note that Sox10 is sparsely expressed and was of minor importance in both the original manuscript and the current, revised one. Raw expression levels, clipped at 8, are shown.

Finally, in addition to the marker genes identified in the Seurat-style analysis, we have made a list of well-known developmental genes from the literature and analyzed their presence in SCENIC regulons, either as transcription factor or as target gene. The results are summarized in Author response image 10. below. All well-known TFs appear in our analysis, though not always at the sufficient confidence level of 0.8; some of them are identified as having regulatory roles, others are simply recognized as target genes. We now comment specifically on confidence levels in the corresponding figure legends (e.g. Figure 4 —figure supplement 2, Figure 5—figure supplement 2) of the revised manuscript.

**Author response image 10. sa2fig10:** Presence of genes known for their role during OB development in regulons of SCENIC. In the left panel, columns are: developmental gene; presence as target gene in given regulon with max confidence cutoffs. Any regulon with a confidence cutoff <0.5 is ignored. In the right panel: developmental gene; found as regulon at max confidence cutoff. Any regulon found with a cutoff <0.4 is ignored.

8) The sub clustering of projection neurons from single nucleus RNA-Seq data is elegant and well-validated with convincing fluorescent in situ hybridization (FISH) data. However, some sort of quantification of overall labeling and overlap between different FISH probes would be useful. Related, it would be broadly beneficial to the field to provide such quantification, particularly for choosing cell-type specific markers in the future.

Thank you for this suggestion. We have now quantified the overlap of the expression of marker genes detected by smFISH in mitral and tufted cell types. We report the results in the revised Figure 3 —figure supplements 1R and 2L. For each image, we first counted the number of cells positive for the less abundant probe (probe A), on average 60 cells for each probe. We then counted amongst probe A-positive cells how many cells also expressed the second probe (probe B). We report overlap as the percentage of probe Apositive cells also expressing probe B. For mitral cell markers we only considered cells along the mitral cell and internal plexiform layers. For tufted cell markers we only considered cells in the external plexiform and glomerular layers. We considered a cell positive for a given smFISH probe if more than 5 dots/cell were present, and if the dots were located within the nucleus and perinuclear compartment of the cell, visualized by DAPI counterstain.

9) Although compelling, the authors may be somewhat stretching the extent to which the gene regulatory network (GRN) clustering recapitulates the transcriptomic clustering. To strengthen this, it may be useful to include some sort of quality measure for GRN clustering to evaluate if class changes are truly meaningful. This might be achieved by clustering based only on the GRN analysis which is not constrained by the originally identified cell classes. This seems important given that the authors use the rationale that there is heterogeneity of active GRNs within transcriptomically identified clusters to justify the sub clustering, and point out that gradual transitions may exist in active GRNs between clusters. This seems to be a fundamental distinction – whether the GRNs functionally define cell types or whether they are a separate axis along which to evaluate cell identity.

Thank you for pointing out this source of confusion. Triggered by this suggestion, we have substantially revised the comparison between transcriptomic and GRN clustering and we refer to comment 3 for the detailed answer. Indeed we have followed the advice of the reviewer and performed Louvain clustering in both transcriptomic and GRN space, and then compared these clusters. Summarizing, results are consistent: cells that are close to each other (i.e. similar) in one space remain similar in the other. Moreover, the mismatches between transcriptomic and GRN clusters occur where we had previously identified heterogeneity and gradual transitions (see Comment 32 for details on gradients).

Regarding the last statement, whether GRNs define cell types or are a separate axis along which one may evaluate cell identity, a mixed picture emerges. Some of our cell types are rather welldefined in both transcriptomic and GRN space, e.g. ET4 and ET2. Contrastingly, while the cell types ET1, T1, and M2 are distinct in transcriptomic space, in GRN space they form a single heterogeneous collective with gradual transitions (see Comment 3, Figure 3.1). On top of that, one may even argue that in GRN space ET1 splits into two regions, one captured by ET4 and the other by T1.

If we speculate on how a GRN is translated into distinct phenotypic outcomes in the context of M/T projection neurons, we might propose (i) that ET1 is a cell type originating from two distinct GRNs, and that (ii) T1 and M2 translate gradients into distinct cell types akin to Wolpert’s French flag (Wolpert 1969, J. Theor. Biol.).

We now discuss this new analysis and its implications on p. 11 l. 274-281 and p. 12 /13 of the revised manuscript, relating to the Figure 4—figure supplement 1.

10) The methods and description of the simulated nuclei approach are somewhat under-described. There are multiple manipulations to the bulk sequencing gene expression (expanding the bulk sequencing data into many simulated nuclei by resampling the counts for regulon/target genes, transforming the simulated counts into Pearson residuals before performing PCA and multiple rounds of classification), but the rationale for and the effects of each of these transformations are relatively unclear. Comparisons to existing algorithms that can deconvolve bulk RNA sequencing data (e.g. Frishberg 2019 Nat Methods, Newman 2019 Nat Biotechnology), would also be useful.

The simulated nuclei approach described here is similar to existing deconvolution methods in its combined use of bulk RNA-seq and scRNA-seq ( e.g. Frishberg Nat Meth 2019). However, our method is unique in its deployment of outside biological information about transcription factor binding sites and gene regulatory networks to harness the relationships between genes to aid the deconvolution. The Methods section has been updated (p. 29-30 in the revised manuscript) with more detailed explanation, relevant portions of which we present below:

“Integration of single nucleus and bulk RNA deep sequencing data using regulon-based simulations

Nuclei were simulated from bulk RNA deep sequencing data using a weighted random sampling of regulons with replacement. A regulon’s relative weight corresponded to the prevalence of its transcription factor in the given bulk RNA-seq sample. Each time a regulon was selected, the counts for its transcription factor and all its target genes increased by one. […]Simulated nuclei predicted to be projection neurons by all 1000 classifiers were designated as putative simulated projection neurons and selected for further analysis. Similarly, these putative simulated projection neurons were integrated with snRNA-seq mitral cells using sctransform. 1000 LDA classifiers were trained to classify simulated nuclei as AON-projecting or PCx-projecting based on values for the 30 top principal components from the sctransform integration. Each classifier was trained on 75% of the simulated projection neurons and tested on the other 25%, with accuracy evaluated using the Jaccard index. Each classifier was then applied to snRNA-seq mitral cells.”

[Editors' note: further revisions were suggested prior to acceptance, as described below.]

Essential Revisions:All the reviewers commend the authors' thorough and thoughtful revisions. There are, however, some points that we would like you to address before publication of this work. You can find these points in the individual reviewers' comments. We expect that modifications of text will be sufficient to address these points. We hope that you can address these points within two weeks.Reviewer #1 (Recommendations for the authors):The authors have addressed many critiques of the reviewers quite thoroughly. Furthermore, they have performed new single-nucleus RNA-sequencing of mitral cells based on their projections into specific sites. The data provide some support of their previous predictions that specific mitral types have preference for specific target sites.

Thank you very much for the support and encouragement.

While I applaud the authors' effort to perform additional single nucleus RNAseq of mitral cells based on their projection to either PCx or AON, the numbers of recovered cells are quite small-they did not report any data on AON-projecting ones because the cell number is too small. Of the 57 cells from the sn-PCx data as projection neurons 44 were classified as M2 cells while 6 were classified as M1 cells. Thus, the heading "Targeted snRNA-seq validates predictions of selective connectivity for molecularly defined mitral cell types" is an overstatement: either the prediction is not completely accurate, or the selectivity is not absolute, or both. This heading and similar statements throughout the text need to be toned down.

Thank you for this suggestion. We have now toned down the corresponding heading on p17 of the revised manuscript (to “preferential connectivity”) as well as abstract, and p19 and p21 of the discussion to address this point.

Reviewer #2 (Recommendations for the authors):This a thoughtful response to all previous reviewers with extensive new data analyses, figure modifications, and clarifications to the text. This is a vastly improved manuscript, and a timely contribution to the field. I commend the authors on a nice revision and fully endorse publication at eLife.

Thank you for your support and encouragement.

Reviewer #3 (Recommendations for the authors):I commend the authors for performing a thorough revision of the paper that addressed our main concerns, especially for the validation data shown in Figure 7 and for the additional details regarding both clustering and the imputation of single nucleus transcriptomes. This paper will serve as an important catalog of molecular diversity in the bulb. I have one experimental request.

Thank you for your suggestions and support.

The experimental request was made in our initial review but was not addressed – unless I missed it (and I may indeed have done so), the authors did not validate the expression of their three key hub genes in MT cells (Taf1, Bclaf1, Pbx3) by either in situ or by showing a UMAP plot describing their expression in the relevant cell types – this is really critical for believing the analysis shown in Figure 5 and should be provided.

Apologies for not more clearly addressing this point in the revision. We now provide UMAP plots as suggested in the revised Figure 5 —figure supplement 1 and describe these on p14 and discuss them on p21.